# Soluble TREM2 ameliorates tau phosphorylation and cognitive deficits through activating transgelin-2 in Alzheimer's disease

Xingyu Zhang[1,2], Li Tang[1,2], Jiaolong Yang[1], Lanxia Meng[1], Jiehui Chen[1], Lingyan Zhou ◉[1], Jiangyu Wang[1], Min Xiong[1] & Zhentao Zhang ◉[1]✉

Triggering receptor expressed on myeloid cells 2 (TREM2) is a transmembrane protein that is predominantly expressed by microglia in the brain. The proteolytic shedding of TREM2 results in the release of soluble TREM2 (sTREM2), which is increased in the cerebrospinal fluid of patients with Alzheimer's disease (AD). It remains unknown whether sTREM2 regulates the pathogenesis of AD. Here we identified transgelin-2 (TG2) expressed on neurons as the receptor for sTREM2. The microglia-derived sTREM2 binds to TG2, induces RhoA phosphorylation at S188, and deactivates the RhoA-ROCK-GSK3β pathway, ameliorating tau phosphorylation. The sTREM2 (77-89) fragment, which is the minimal active sequence of sTREM2 to activate TG2, mimics the inhibitory effect of sTREM2 on tau phosphorylation. Overexpression of sTREM2 or administration of the active peptide rescues tau pathology and behavioral defects in the tau P301S transgenic mice. Together, these findings demonstrate that the sTREM2-TG2 interaction mediates the cross-talk between microglia and neurons. sTREM2 and its active peptide may be a potential therapeutic intervention for tauopathies including AD.

Alzheimer's disease (AD) is the most common neurodegenerative disease and the most common cause of dementia. Pathologically, AD is characterized by extracellular aggregates composed of amyloid-β (Aβ) and intraneuronal neurofibrillary tangles (NFTs) composed of hyperphosphorylated tau[1]. Under physiological conditions, tau regulates microtubule dynamics and intracellular trafficking[2]. During the onset of AD, tau is hyperphosphorylated, which interferes with its physiological function[3], and promotes the formation of insoluble aggregates[4]. Thus, hyperphosphorylation is believed to be the most important post-translational modification that induces tau pathology in AD. Despite extensive studies, the molecular mechanisms that regulate tau phosphorylation during the onset and progression of AD are poorly understood.

Microglia plays a complex role in the brain and is implicated in the development of AD-related pathology. Interestingly, neurons containing NFTs are often surrounded by activated microglia in the AD brain[5,6]. Microglia sense neuronal activity and modulate neuronal function through the cell-cell signaling pathways[7]. It has been reported that microglia promote tau pathology via exosome secretion, while depletion of microglia or inhibition of exosome synthesis halts tau propagation[8]. Furthermore, inhibition of the NLRP3 inflammasome in microglia reduced tau pathology[9]. However, there are also studies showing that activated microglia mitigate tau seeding and spreading[10]. Moreover, interleukin-3 (IL-3) programs microglia to ameliorate the pathology of AD[11]. These observations indicate that distinct microglial activation phenotypes may contribute differently to tau pathology.

[1]Department of Neurology, Renmin Hospital of Wuhan University, Wuhan 430060, China. [2]These authors contributed equally: Xingyu Zhang, Li Tang. ✉e-mail: zhentaozhang@whu.edu.cn

Understanding how microglia regulate tau pathology in neurons will be key to developing effective therapies for AD.

Triggering receptor expressed on myeloid cells 2 (TREM2) is a transmembrane receptor expressed in myeloid cells, including microglia in the brain. Genetic studies have indicated that rare variants in TREM2 increase the risk of AD by ~2- to 3-folds[12,13]. Previous studies found that TREM2 regulates the response of the innate immune system

to Aβ[14]. However, the role of TREM2 in tau pathology remains controversial. Several studies found that silencing of brain TREM2 exacerbates tau pathology[15–17] and overexpression of TREM2 ameliorates tau pathology[17], but other studies found that deletion of TREM2 did not affect tau phosphorylation[18,19], while TREM2 activation exacerbated tau pathology[20]. Thus, the effect of TREM2 on tau pathology deserves further investigation.

**Fig. 1 | sTREM2 attenuates tau hyperphosphorylation in vitro.** Western blots (**a**) and quantification (**b**) tau phosphorylation in HEK293-Tau cells treated with Fc, heat-inactivated Fc-sTREM2 (40 nM), or Fc-sTREM2 (40 nM) for 24 h (mean ± s.e.m.; one-way ANOVA, n = 5 independent experiments). Immunostaining (**c**) and quantification (**d**) of p-Tau S202 in HEK293-Tau cells treated with Fc (40 nM), heat-inactivated Fc-sTREM2 (40 nM), or Fc-sTREM2 (40 nM) for 24 h. Scale bar, 25 μm (mean ± s.e.m.; one-way ANOVA, n = 10 images from 5 independent experiments). p-Tau S202 immunostaining (**e**) and quantification (**f**) of HEK293-Tau cells treated with different concentrations of sTREM2. The control group was treated with Fc. Scale bars, 25 μm (mean ± s.e.m.; one-way ANOVA, n = 10 images from 5 independent experiments; compared with the control group). Western blots (**g**) and quantification (**h**) showing the phosphorylation of tau and GSK3β in HEK293-Tau cells treated with different concentrations of sTREM2. The control group (sTREM2 = 0 nM) was treated with Fc (mean ± s.e.m.; one-way ANOVA, n = 4 independent experiments; compared with the control group). p-Tau S202 and p-tau S396 immunostaining (**i**) and quantification (**j**) of primary neurons derived from tau P301S mice treated with different concentrations of sTREM2. Scale bar, 20 μm. The fluorescence intensity in the field was normalized to the control group (mean ± s.e.m.; one-way ANOVA, n = 10 images from 5 independent experiments; compared with the control group), *$P < 0.05$, **$P < 0.01$, ***$P < 0.001$.

TREM2 is cleaved by α-secretase (ADAM10 and ADAM17) after the H157 residue, releasing soluble TREM2 (sTREM2)[21,22]. The levels of sTREM2 in the cerebrospinal fluid (CSF) are increased in AD subjects, especially in the early stage of AD[23,24]. Higher levels of sTREM2 are associated with diminished cognitive decline[25,26], suggesting that sTREM2 may play a beneficial role in AD. Furthermore, sTREM2 has been found to be localized on the surface of neuron[27], indicating that it may act as a mediator of microglia-neuron communication. However, it remains unknown whether and how sTREM2 regulates tau pathology.

Here, we sought to illustrate the potential effects of sTREM2 on tau phosphorylation in neurons, and to identify the neuronal receptor that mediates the effect of sTREM2. We found that sTREM2 selectively binds to transgelin-2 (TG2), inhibits the downstream RhoA-ROCK pathway, and attenuates tau hyperphosphorylation. Overexpression of sTREM2 attenuates tau pathology and cognitive impairments in tau P301S transgenic mice. We also identified that sTREM2 (77–89) is the minimal sequence of sTREM2 that activates TG2. This active peptide mimics the protective effect of sTREM2 both in vitro and in vivo.

## Results

### sTREM2 attenuates tau hyperphosphorylation in vitro

To explore the effects of sTREM2 on tau phosphorylation, we constructed a HEK293 cell line stably overexpressing GFP-Tau (HEK293-Tau cells), and treated the cells with Fc-vector, heat-inactivated sTREM2, or sTREM2 (40 nM). The purity and concentrations of sTREM2 were verified (Supplementary Fig. 1a, b). Interestingly, sTREM2 reduced tau phosphorylation at S202, S396, T181, and S404 residues, with S202 and S396 being the most dramatically altered sites (Fig. 1a–d, Supplementary Fig. 1c). Thus, we focused on the S202 and S396 residues. Interestingly, the inhibitory effect of sTREM2 on tau phosphorylation was dose-dependent (Fig. 1e–h, Supplementary Fig. 1d, e). This phenomenon was verified in primary neurons cultured from tau P301S transgenic mice (Fig. 1i, j). In addition, TUNEL assays and LDH release assays revealed that sTREM2 protected the primary neurons of tau P301S mice from cell death (Supplementary Fig. 1f–h).

Since the S202, S396, T181, and S404 sites of tau are the targets of glycogen synthase kinase-3β (GSK3β), we further detected the activity of GSK3β in the presence of sTREM2. We found that sTREM2 dramatically decreased GSK3β phosphorylation at Y216 in a concentration-dependent manner, indicating decreased GSK3β activity induced by sTREM2 (Fig. 1g, h). Additionally, the conditioned medium from BV2 cells overexpressing sTREM2 reduced the phosphorylation of tau in HEK293-Tau cells, while depletion of sTREM2 using an anti-TREM2 antibody abolished the inhibitory effect on tau phosphorylation (Supplementary Fig. 2a–c). Furthermore, purified sTREM2 attenuated tau phosphorylation and GSK3β activation induced by Aβ oligomers (Supplementary Fig. 3a, b). Together, these results suggest that sTREM2 attenuates tau hyperphosphorylation by inhibiting GSK3β.

### sTREM2 interacts with TG2

To explore whether microglia-derived sTREM2 inhibits tau phosphorylation by interacting with receptors on the neuronal surface, we purified recombinant Fc-tagged sTREM2 and performed affinity purification experiments to pull down interacting proteins from the cell membrane fraction extracted from SH-SY5Y cells. The purity of the cell membrane fraction was confirmed by Western blotting (Supplementary Fig. 4a). The candidate receptors of sTREM2 were analyzed by mass spectrometry (Fig. 2a). We identified 17 proteins that may interact with sTREM2 (Supplementary Table 1), among which transgelin-2 (TG2) (Fig. 2b) was the only protein that has been reported to act as a cell-surface receptor[28]. To further confirm that sTREM2 binds TG2 in the neuronal cell membrane, we extracted the cell membrane fraction of primary neurons from WT mice and tau P301S mice, incubated them with sTREM2, and performed affinity purification-mass spectrometry. Consistently, sTREM2 was found to interact with TG2 in the neuronal membrane fraction (Fig. 2c, Supplementary Tables 2, 3).

Furthermore, TG2 and sTREM2 colocalized with neuronal marker (MAP2) and glial markers (GFAP and Iba1) in hippocampal brain sections from AD patients and tau P301S mice. The specificity of the anti-sTREM2 antibody was confirmed by Western blotting and immunohistochemistry (Supplementary Fig. 4b–d). We found that sTREM2 colocalized with TG2 in both neurons and glial cells (Fig. 2d–f, Supplementary Fig. 4e). In the cell-surface binding assay, Fc-tagged sTREM2, but not Fc itself, attached to the surface of HEK293 cells overexpressing TG2 but not that of control cells (Fig. 2g, h). Interestingly, knockdown of TG2 attenuated the attachment of sTREM2 to neurons (Supplementary Fig. 4f).

Furthermore, the cell membrane proteins of SH-SY5Y cells or primary neurons were extracted, incubated with purified Fc-sTREM2, and then pulled down with protein A beads. TG2 was found in the membranous fraction of SH-SY5Y cells and primary neurons (Fig. 2i, j). Fc-sTREM2, but not Fc itself, interacted with TG2 (Fig. 2i, j). Moreover, we co-transfected His-TG2 and GST-sTREM2 into HEK293 cells. GST pull-down assay also confirmed that sTREM2 binds TG2 (Fig. 2k). Overall, these results demonstrate that sTREM2 interacts with TG2.

### sTREM2 inhibits tau phosphorylation by activating TG2

TG2 has recently been shown to localize at the cellular membrane and act as a receptor to regulate the myosin cytoskeleton of airway smooth muscle cells[29]. To explore whether TG2 is involved in the regulation of tau phosphorylation by sTREM2, we treated HEK293-tau cells with the TG2 agonist TSG12[29]. Interestingly, TSG12 dramatically decreased the levels of p-tau S202 and S396 in HEK293-tau cells (Fig. 3a–c, Supplementary Fig. 5a). Furthermore, TSG12 also inhibited the activation of GSK3β (Fig. 3a, b). Similar results were observed in primary neurons (Fig. 3d, Supplementary Fig. 5b). These results indicate that the TG2 agonist mimics sTREM2 in inhibiting tau phosphorylation. To verify the role of TG2 in the inhibitory effect of sTREM2 on tau phosphorylation, we knocked down TG2 in HEK293-tau cells or primary neurons cultured from tau P301S mice using shRNA and then treated the cells with sTREM2. Interestingly, the effect of sTREM2 on tau phosphorylation and GSK3β activation was abolished in the absence of TG2 (Fig. 3e–h, Supplementary Fig. 5c, d). Similar effect was found in wild-type neurons (Supplementary Fig. 5e). These results indicate that TG2 is required for the inhibitory effect of sTREM2 on tau phosphorylation.

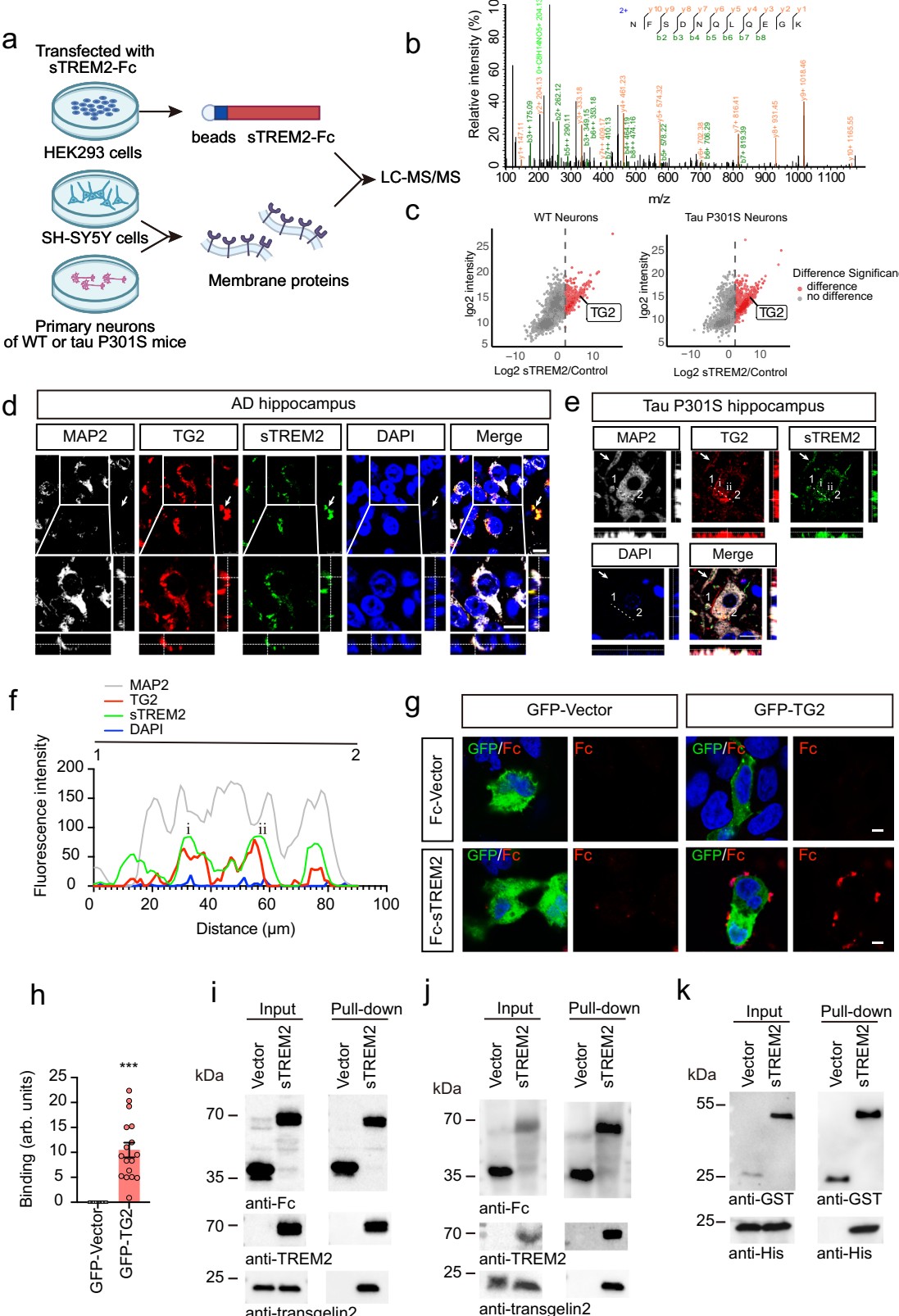

## The RhoA/ROCK pathway is involved in the regulation of tau phosphorylation by sTREM2

It has been reported that TG2 induces RhoA phosphorylation at S188, thereby inactivating the RhoA-ROCK pathway[29], which is involved in the activation of GSK3β[30] (Fig. 4a). To investigate whether sTREM2 activates the TG2-Rho-ROCK pathway, we treated SH-SY5Y cells with sTREM2 and tested the levels of GTP-RhoA, which is the active form of RhoA[31], and p-RhoA S188, which is the inactive form of RhoA[32]. We found that sTREM2 treatment decreased the levels of RhoA-GTP and elevated the levels of p-RhoA S188 (Supplementary Fig. 6a, b). These results suggest that sTREM2 inhibits the activation of RhoA. To further explore the role of the RhoA-ROCK pathway in the inhibitory effect of

**Fig. 2 | sTREM2 interacts with TG2. a** Cartoon illustrating the AP-MS workflow. LC-MS/MS, liquid chromatography-tandem mass spectrometry. **b** MS/MS spectrum showing the identification of TG2. **c** Volcano plot of neuronal cell membrane proteins of WT mice (left) or tau P301S mice (right) that interact with Fc-sTREM2. The cutoff threshold was set at log$_2$FC > 2. **d** The co-localization of neuronal marker (MAP2), TG2, and sTREM2 in the hippocampus of AD patients. Arrows indicate TG2 expression in non-neuronal cells. Scale bars, 10 μm. $n = 5$ independent experiments. **e** The co-localization of MAP2, TG2, and sTREM2 in the hippocampus of tau P301S mice. The arrows indicate the colocalization in branches. Scale bars, 10 μm. $n = 5$ independent experiments. (**f**) The intensity trace (offset white line) of (**e**) is potted.

**g** Confocal images of immunostaining for Fc-sTREM2 binding to GFP-TG2. Scale bar, 5 μm. **h** Quantification of immunostaining in (**g**). The binding was calculated as the area of Fc immunostaining relative to the cell area (mean ± s.e.m.; Two-tailed t-test, $n = 17$ images per group; compared with the control group, ***$P < 0.001$). arb. units, arbitrary units. TG2 is found in the membranous fraction of SH-SY5Y cells (**i**) and primary neurons of WT mice (**j**). Pull-down assay using protein A beads indicated that membrane TG2 binds Fc-sTREM2. $n = 5$ independent experiments. **k** GST pull-down assay verified the interaction between GST-sTREM2 and His-TG2. $n = 5$ independent experiments.

tau phosphorylation by sTREM2, we treated HEK293-tau cells with a RhoA inhibitor (Tat-C3) and ROCK inhibitor (Y-27632), respectively. Both Tat-C3 and Y-27632 mimicked the inhibitory effect of sTREM2 on tau phosphorylation and GSK3β activation (Fig. 4b–e). In contrast, exposure to the RhoA activator abolished the effect of sTREM2 on tau phosphorylation and GSK3β activation (Fig. 4f, g). To further confirm that sTREM2 attenuates tau phosphorylation by inhibiting the RhoA-ROCK pathway, we used siRNA to knock down the expression of RhoA and ROCK, respectively. Both siRNAs mimicked the effect of sTREM2 in reducing tau phosphorylation (Fig. 4h–k). The effect of the RhoA-ROCK pathway on tau phosphorylation was further verified by immunofluorescence staining (Supplementary Fig. 6c, d).

To confirm that sTREM2 inhibits tau phosphorylation by inducing the phosphorylation of RhoA at S188, we transfected cells with wild-type RhoA and the phosphorylation-resistant mutant (RhoA S188A) in an endogenous RhoA-null background. The S188A mutant blocked the inhibitory effect of sTREM2 on tau phosphorylation (Supplementary Fig. 7a–c). These results confirm that sTREM2 inhibits tau phosphorylation by blocking the RhoA-ROCK pathway. Furthermore, we detected the TG2-RhoA-ROCK-GSK3β pathway in AD patients. We found that the levels of TG2 and RhoA phosphorylation at S188 were decreased in AD patients, whereas the level of GSK3β phosphorylation at Y216 was increased (Supplementary Fig. 8a–c), indicating that the TG2-RhoA-ROCK-GSK3β pathway is impaired in AD.

### sTREM2 attenuates tau hyperphosphorylation and memory loss in tau P301S mice

To further investigate whether sTREM2 regulates tau phosphorylation in vivo, we injected AAVs encoding EGFP-sTREM2 or EGFP vector into the hippocampus of 3-month-old tau P301S mice. The mice were analyzed at 7 months of age. We observed strong expression of EGFP and EGFP-sTREM2 in the hippocampus (Supplementary Fig. 9a, b). As expected, injection of AAV-EGFP-sTREM2 increased the concentrations of sTREM2 in the hippocampus (Supplementary Fig. 9c). The phosphorylation of tau was ameliorated in the hippocampus of tau P301S mice injected with AAV-sTREM2 compared with tau P301S mice injected with AAV-vector. The activity of GSK3β in tau P301S mice was also decreased in the presence of sTREM2 (Fig. 5a–c). Furthermore, electron microscopy showed that sTREM2 protected against the loss of hippocampal synapses in tau P301S mice (Fig. 5d). Golgi staining revealed that the density of dendritic spines was higher in mice injected with AAV-sTREM2 than in mice injected with AAV-vector (Fig. 5e). The loss of inhibitory and excitatory synapses in the hippocampus of tau P301S mice was attenuated when sTREM2 was overexpressed (Supplementary Fig. 9d–g).

We further performed behavioral tests to investigate the effect of sTREM2 on cognitive function. In the water maze test, tau P301S mice injected with AAV-EGFP-sTREM2 showed better memory, as shown by a shorter time to find the platform during the training phase (Fig. 5f, g) and longer time in the target quadrant during the probe test (Fig. 5h) than tau P301S mice injected with AAV-vector. Consistently, the mice injected with AAV-sTREM2 spent more time in the new arm during the Y-maze test (Fig. 5i). Hippocampal long-term potentiation (LTP) is considered to be the basis of learning and memory. The

electrophysiological study found that the LTP of fEPSCs was elevated in tau P301S mice overexpressing sTREM2 compared with tau P301S mice injected with AAV-vector (Fig. 5j, k). These results indicate that sTREM2 reverses tau pathology, synaptic dysfunction, and cognitive impairment in tau P301S mice.

### TG2 knockdown abrogates the effect of sTREM2 in vivo

To further verify the role of TG2 in the progression of tau pathology in vivo, we knocked down the expression of TG2 by injecting AAV-EGFP-shRNA-TG2 into the hippocampus of 3-month-old tau P301S mice. Mice were sacrificed at 7 months of age. The expression of EGFP is shown in Supplementary Fig. 10a. AAV-shRNA-TG2 successfully knocked down the expression of TG2 in the hippocampus (Fig. 6b, Supplementary Fig. 10b). In mice injected with control AAV-shRNA (sh-NC), overexpression of sTREM2 attenuated tau hyperphosphorylation in the hippocampus, whereas the inhibitory effect of sTREM2 on tau phosphorylation was abolished when TG2 was knocked down. Furthermore, sTREM2 deactivated the RhoA-ROCK-GSK3β pathway, which was abolished in mice injected with sh-TG2 (Fig. 6a, b). These results suggest that the effect of sTREM2 on tau phosphorylation depends on TG2. Furthermore, electron microscopy showed that sTREM2 protected against the loss of hippocampal synapses in tau P301S mice injected with sh-NC, but not in tau P301S mice injected with sh-TG2 (Fig. 6c). Furthermore, knockdown of TG2 attenuated the protective effect of sTREM2 on the densities of inhibitory and excitatory synapses in the hippocampus of tau P301S mice (Supplementary Fig. 10c–f). Thus, these results point to a critical role of TG2 in mediating the beneficial effect of sTREM2.

### A short peptide of sTREM2 suppresses tau hyperphosphorylation in vitro

Given the beneficial effect of sTREM2, we tried to find an active peptide within sTREM2 to mimic the effect of sTREM2 on tau phosphorylation. We designed four fragments of sTREM2, including sTREM2 (amino acids 20–46, 47–76, 77–105, and 106–136) (Fig. 7a), and tested their ability to bind TG2. The sTREM2(77–105) fragment was sufficient to bind with TG2 (Fig. 7b). We further synthesized 4 short peptides based on sTREM2(77-105) (Fig. 7a). Strikingly, peptide 1 (amino acids 77–89) dramatically inhibited tau phosphorylation, similar to sTREM2 (Fig. 7c–f, Supplementary Fig. 11a, b). Furthermore, this peptide inhibited the activation of the RhoA-ROCK-GSK3β pathway in a concentration-dependent manner (Supplementary Fig. 11c–f). Furthermore, peptide 1 also attenuated tau phosphorylation and GSK3β activation induced by Aβ oligomers (Supplementary Fig. 12a, b). In addition, we found that FITC-peptide 1 attached to neurons, which was attenuated when TG2 was knocked down (Supplementary Fig. 4f). These results demonstrate that peptide 1 is sufficient to bind neurons via TG2 and inhibit tau phosphorylation.

### The active sTREM2 peptide suppresses tau pathology in tau P301S mice

We next assessed the effect of peptide 1 on tau hyperphosphorylation in tau P301S mice. We conjugated peptide 1 or the

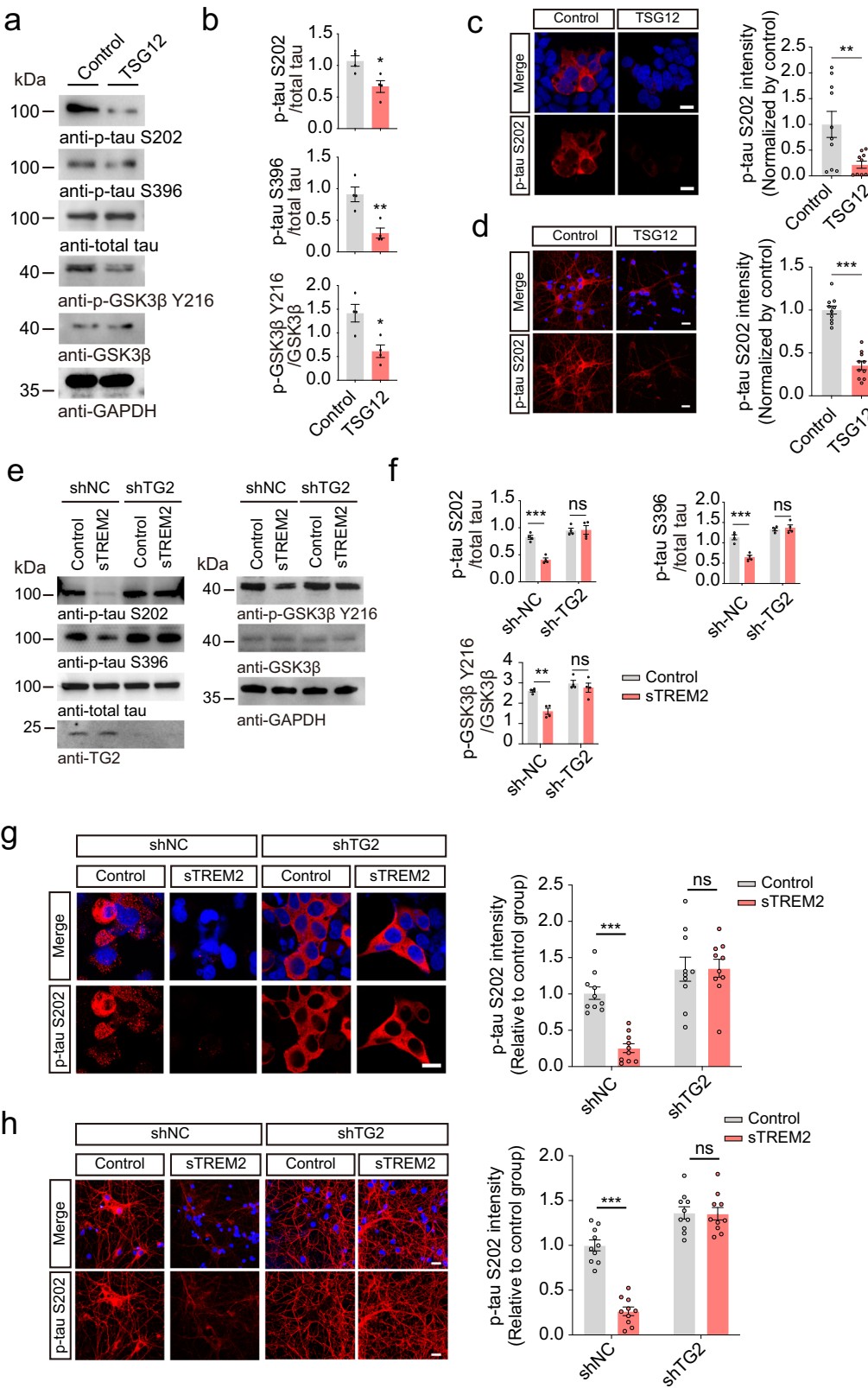

reversed peptide (89–77) with a modified HIV transactivator of transcription (Tat) peptide to facilitate their delivery into the brain. To examine the ability of the peptides to cross the blood-brain barrier and their stability in the brain, FITC-labeled peptides (10 mg/kg) were injected intraperitoneally (i.p.) into the mice. The fluorescence in brain sections was observed at 1, 3, 5, and 7 days after injection (Supplementary Fig. 13a, b). We found that the

peptide was brain-permeable and detectable in different brain areas, including the hippocampal CA1, CA3, DG, fimbria, hypo-thalamus, entorhinal cortex, and amygdala, until 7 days after injection (Fig. 8a, Supplementary Fig. 13c). In addition, we found that FITC-peptide 1 colocalized with TG2 and MAP2 (Supple-mentary Fig. 13d), suggesting that peptide 1 interacts with TG2 on neurons.

**Fig. 3 | sTREM2 inhibits tau phosphorylation by activating TG2. a** Western blots showing the phosphorylation of tau and GSK3β in HEK293-Tau cells treated with vehicle or TSG12 (25 μM). **b** Quantification of the immunoreactivity in (**a**) (mean ± s.e.m.; Two-tailed *t*-test, *n* = 4 independent experiments; compared with the control group). **c** p-Tau S202 immunostaining in HEK293-Tau cells treated with vehicle or TSG12 (mean ± s.e.m.; two-way ANOVA, *n* = 10 images from 5 independent experiments). Scale bars, 15 μm. **d** p-Tau S202 immunostaining in vehicle- or TSG12-treated (25 μM) primary neurons derived from tau P301S mice (mean ± s.e.m.; Two-tailed *t*-test, *n* = 10 images from 5 independent experiments). Scale bars, 20 μm. **e** HEK293-Tau cells were pretreated with sh-NC or sh-TG2, and then exposed to Fc-sTREM2 (20 nM) or control (Fc, 20 nM) for 24 h. Shown is the phosphorylation of

tau and GSK3β. **f** Quantification of the immunoreactivity in (**e**) (mean ± s.e.m.; two-way ANOVA, *n* = 4 independent experiments). **g** p-Tau S202 immunostaining and quantification of HEK293-Tau cells pretreated with control shRNA (sh-NC) or sh-TG2 and then exposed to Fc-sTREM2 (20 nM) or control (Fc, 20 nM) for 24 h. Scale bars, 20 μm (mean ± s.e.m.; two-way ANOVA, *n* = 10 images from 5 independent experiments). **h** p-Tau S202 immunostaining and quantification of primary neurons derived from tau P301S mice pretreated with control shRNA (sh-NC) or sh-TG2 and then exposed to Fc-sTREM2 (20 nM) or control (Fc, 20 nM) for 24 h. Scale bars, 20 μm. (mean ± s.e.m.; two-way ANOVA, *n* = 10 images from 5 independent experiments). *$P < 0.05$, **$P < 0.01$, ***$P < 0.001$.

Thus, we i.p. injected 10 mg/kg Tat-sTREM2 (77–89) or Tat-sTREM2 (89-77) into three-month-old tau P301S mice once every 5 days for 4 months. The mice were analyzed at 7 months of age. Similar to sTREM2, peptide 1 dramatically inhibited tau phosphorylation and the activation of GSK3β and RhoA in the hippocampus (Fig. 8b–d). Peptide 1 also ameliorated the loss of synapses in tau P301S mice (Fig. 8e, f).

In the water maze test, the mice administered peptide 1 spent less time finding the platform during the training phase (Fig. 8g, h) and spent a longer time in the target quadrant during the probe trial than mice administered the reversed peptide (Fig. 8i). In the Y-maze test, the mice administered peptide 1 spent more time in the new arm (Fig. 8j). These results indicate that peptide 1 improves the learning and memory function of tau P301S mice. Moreover, the electro-physiological study also found that the LTP of fEPSCs was elevated in mice injected with peptide 1 compared with mice injected with the reversed peptide (Fig. 8k, l), which suggests that peptide 1 ameliorates synaptic dysfunction in tau P301S mice. Overall, peptide 1 mimics sTREM2 to ameliorate tau pathology, synaptic dysfunction, and cognitive impairment in tau P301S mice.

## Discussion

In the present study, we found that microglia-derived sTREM2 inhibits tau hyperphosphorylation in neurons. We also identified TG2 as a novel neuronal receptor for sTREM2. Consequently, activation of TG2 by sTREM2 inhibits the RhoA-ROCK-GSK3β pathway, ameliorating tau phosphorylation in neurons. Our observations indicate that the sTREM2-TG2 interaction acts as a novel mechanism to halt the progression of tau pathology. Furthermore, we defined a 13-amino acid peptide, sTREM2 (77-89), which is the minimal active sequence of sTREM2 that mimics the effect of sTREM2 and plays a protective role in tau pathology.

The role of TREM2 in tau pathology remains controversial. Several studies have shown that silencing brain TREM2 exacerbates tau pathology in tau P301S mice or human tau transgenic mice[15–17], while overexpression of TREM2 ameliorates tau pathology[17,33]. In contrast, other studies found that deletion of TREM2 did not affect tau phosphorylation in tau P301S mice[18] and in pR5-183 mice[19], while TREM2 activation exacerbated Aβ-associated tau pathology in a mouse model of amyloidosis[20]. The contradictory effect may be due to different mouse backgrounds and different ways to knock down or overexpress TREM2. These studies indicate that TREM2 plays an important but mysterious role in tau pathology.

Shedding of TREM2 by α-secretase results in the release of sTREM2, which may act as a mediator of microglia-neuron interactions. The level of sTREM2 is increased in the CSF from AD patients[23,24] and peaks at the early symptomatic stage of the disease[24]. Higher levels of sTREM2 are associated with diminished cognitive decline[25,26], Aβ deposition, and tau pathology, suggesting that sTREM2 may play a beneficial role in AD. However, the effect and mechanisms of sTREM2 on tau phosphorylation remain unclear. Tau phosphorylation at S202, S396, S404, T181, and T231 residues is involved in the pathogenesis of AD[34–37]. We found that sTREM2

potently inhibits the phosphorylation of tau at the S202, S396, T181, and S404 residues, but not at T231. These results indicate that pathways unrelated to sTREM2 are involved in tau phosphorylation at T231.

We focused on tau phosphorylation at the S202 and S396 residues since they are altered by sTREM2 and are implicated in AD pathogenesis. sTREM2 also ameliorates tau hyperphosphorylation in tau P301S mice. These results are consistent with previous studies showing that the depletion of TREM2 exacerbates neuronal tau hyperphosphorylation[15–17]. Our results demonstrate that not only does TREM2 expressed on the cell membrane play a role, but the secreted form of TREM2 also directly inhibits tau phosphorylation.

sTREM2 is secreted by microglia. How does it affect the phosphorylation of tau in neurons? To answer this question, we performed mass spectrometry to identify the receptor of sTREM2 and found that sTREM2 binds TG2. Among all of the identified proteins, only TG2 was reported as a receptor for extracellular ligands and induces RhoA phosphorylation[28,29]. TG2 can be S-palmitoylated and may serve as a lipid-anchored protein in the cell membrane[38,39]. Consistently, converging lines of evidence show that TG2 is expressed in cell membranes[38,40,41]. Here we found that TG2 is expressed in the cell membrane of SH-SY5Y cells and primary neurons and functions as a receptor. Although the mRNA level of TG2 in neurons is relatively low as detected by RNAseq[41,42], a high-resolution mass spectrometry-based proteomics for in-depth analysis of the major brain regions and cell types identified that TG2 is evenly expressed in neurons and glial cells[43]. We found that TG2 is expressed in both neurons and glial cells, and colocalizes with sTREM2 (Fig. 2d–f, Supplementary Fig. 4e). It has been reported that sTREM2 modulates microglial functions[44], but it remains unknown whether sTREM2 also regulates the activity of astrocytes. In our study, we focused on the effect of sTREM2 on neurons. We extracted the cell membrane fractions of primary neurons from WT mice and tau P301S mice and performed affinity purification-mass spectrometry. We found that sTREM2 interacts with TG2 in the cell membrane of neurons from both WT mice and tau P301S mice. These results indicate that sTREM2 interacts with TG2 under both basal and AD pathological conditions. How the interaction between sTREM2 and TG2 is modulated during AD deserves further research.

TG2 agonists have been reported to induce RhoA phosphorylation[28,29]. Consistent with the effect of the TG2 agonist, sTREM2 induces RhoA phosphorylation at S188, which deactivates RhoA. Active RhoA binds with several effector proteins, including ROCK, to transmit downstream signals[45]. Aβ has been found to activate the RhoA/ROCK/GSK3β signaling pathway, leading to tau hyperphosphorylation[30]. In contrast to Aβ, sTREM2 binds to TG2 and inhibits the downstream RhoA/ROCK/GSK3β signaling pathway to reduce tau hyperphosphorylation (Fig. 4a). Activation of the RhoA-ROCK pathway abolishes the effect of sTREM2 on tau pathology.

Microglia play a complicated role in the pathophysiology of AD. The communication between microglia and neurons remains poorly defined. In our research, we found that microglia secrete sTREM2,

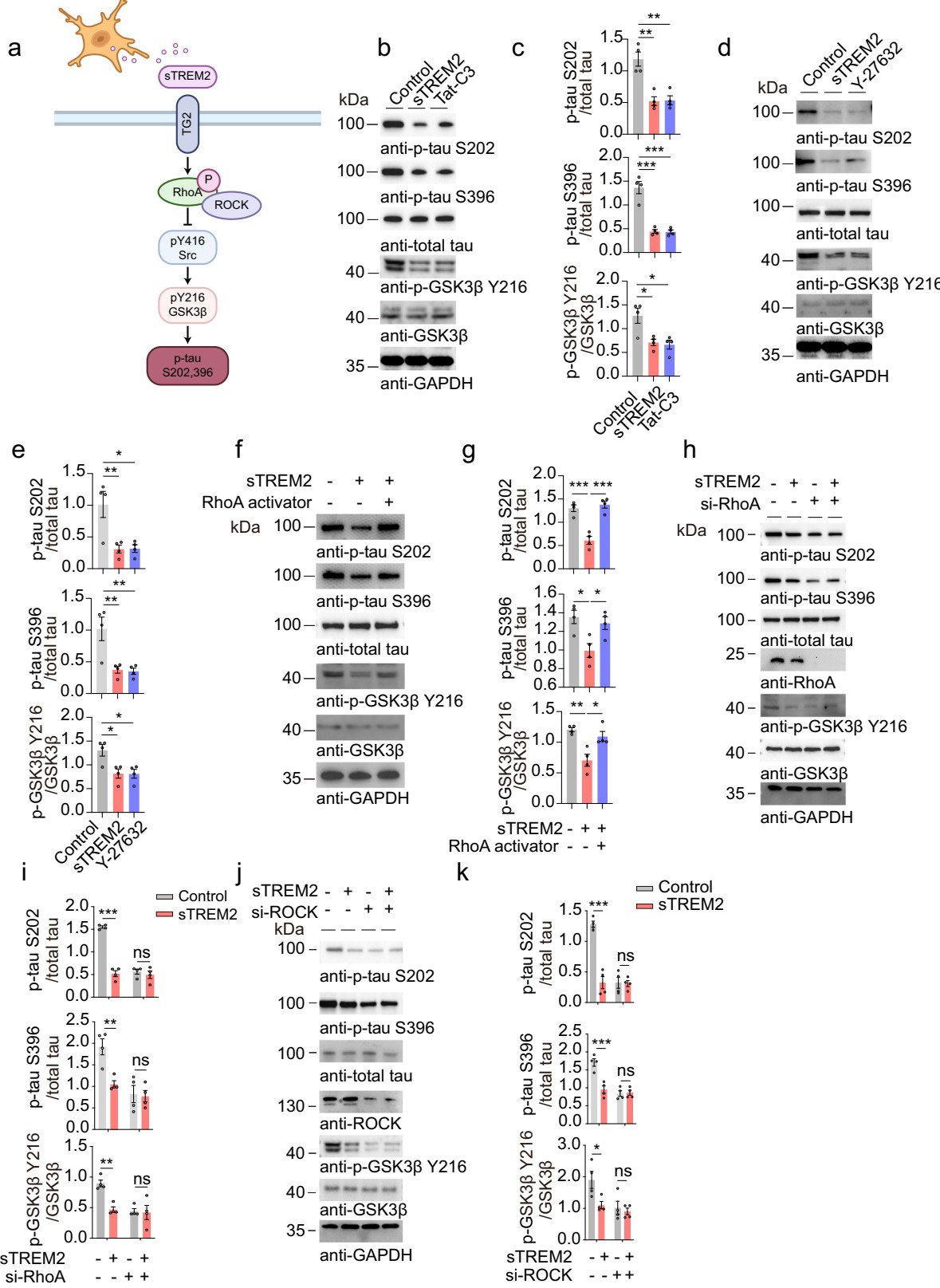

which may play a protective role in tau pathology. We proposed that during the onset of AD, microglia are activated and secrete sTREM2 to protect neurons from tau pathology. Thus, we designed a peptide to mimic the protective effect of sTREM2. This 13-amino acid peptide significantly ameliorates the pathology and behavioral deficits in tau P301S mice. This may provide a potential therapeutic intervention for treating tauopathies including AD.

## Methods

### Ethics statement

The studies were reviewed and approved by the Animal Care and Use Committee of Renmin Hospital of Wuhan University. The post-mortem brain samples were collected with the approved consent of the Xiangya School of Medicine and the Emory Alzheimer's Disease Research Center (ADRC).

**Fig. 4 | sTREM2 attenuates tau phosphorylation by blocking the RhoA/ROCK pathway. a** Schematic illustration of the potential mechanism by which sTREM2 regulates tau phosphorylation: sTREM2 secreted from microglia binds neuronal TG2 and inactivates the RhoA/ROCK/GSK3β pathway, attenuating tau hyperphosphorylation. **b, c** Western blots showing the phosphorylation of tau and GSK3β in HEK293-Tau cells treated with control (Fc, 20 nM), Fc-sTREM2 (20 nM), or Tat-C3 (RhoA inhibitor, 0.1 μg/mL) for 24 h. **d, e** Western blots showing the phosphorylation of tau and GSK3β in HEK293-Tau cells treated with control (Fc, 20 nM), Fc-sTREM2 (20 nM), or Y-27632 (ROCK inhibitor) for 24 h. **f, g** Western blots showing the phosphorylation of tau and GSK3β in HEK293-Tau cells treated with control (Fc, 20 nM), Fc-sTREM2 (20 nM) for 24 h and incubated with RhoA activator (1 U/ml) or solvent for another 30 min. EK293-Tau cells were pretreated with control siRNA (si-NC), si-RhoA (**h, i**), or si-ROCK (**j, k**), and then treated with control (Fc, 20 nM) or Fc-sTREM2 (20 nM). Shown are the phosphorylation of tau and GSK3β (mean ± s.e.m.; one-way ANOVA for (**c, e, g**), two-way ANOVA for (**i, k**), $n = 4$ independent experiments; compared with the control group, \*$P < 0.05$, \*\*$P < 0.01$, \*\*\*$P < 0.001$).

## Mice

Tau P301S mice (stock number: 008169) and wild-type C57BL/6J mice (stock number: 000664) were obtained from the Jackson Laboratory. Animal care and handling were carried out following the Declaration of Helsinki and the guidelines of Renmin Hospital, Wuhan University. The mice were housed under specific pathogen-free (SPF) conditions, following a 14-h light and 10-h dark cycle. They were provided with unrestricted access to food and water. The sample size was estimated by Power and Precision (Biostat). Only male mice were used in this study. The animals were randomized to the different experimental conditions and treatments using the random number generator function (RANDBETWEEN) in Microsoft Excel. The researchers were unaware of the group assignment.

## Human tissue samples

The human brain samples of AD patients and age-matched controls for western blotting were obtained from the Xiangya School of Medicine. The brain tissues for immunostaining were obtained from the Emory Alzheimer's Disease Research Center (ADRC) Brain Bank. The diagnosis of AD was confirmed by the presence of amyloid plaques and neurofibrillary tangles. The age at death of patients ranged between 52 and 70 years old. The post-mortem intervals (PMI) were 2.5–9 h. Brain sections used in this study were prepared from hippocampal tissue. Informed consent for both use of the samples and publication of the identifiers and results was obtained from the subjects or their authorized representatives.

## Cells and treatment

All cell lines were obtained from the American Tissue Culture Collection (ATCC). SH-SY5Y cells are a subclone of the SK-N-SH neuroblastoma cell line, while BV-2 cells are a type of microglial cell derived from C57/BL6 mice. HEK293 cells were used for transient transfections using polyethyleneimine (PEI). The HEK293 cell line stably overexpressing GFP-Tau (HEK293-Tau cells) was established via LV-EF1a-EGFP-tau (1–441 aa) infection and puromycin selection. To investigate the effect of sTREM2 on tau phosphorylation, HEK293-Tau cells were serum-starved for 12 h and treated with Fc, heat-inactivated Fc-sTREM2 (40 nM) or purified Fc-sTREM2 (40 nM) for 24 h. For dose-dependent assays, HEK293-Tau cells or primary neurons from tau P301S were treated with Fc (control) or Fc-sTREM2 (4 nM, 20 nM, 40 nM) for 24 h. To confirm the concentrations of Fc-sTREM2 in the culture medium, the medium was collected for TREM2 ELISA assay immediately after adding sTREM2.

To overexpress sTREM2 in BV2 cells, the cells were infected with AAV-sTREM2. The conditioned medium (CM) was collected 48 h later. To deplete sTREM2 from the CM, the TREM2 antibody (27599-1-AP, Proteintech) was added to the CM at a concentration of 10 μg/ml and incubated for 2 h at 37 °C. Then, the medium was incubated with 100 μl Protein A agarose beads for 6 h at 4 °C. The sample was centrifuged at 400 × g for 5 min, and the supernatant was filtered and harvested. HEK293-tau cells were exposed to sTREM2 CM, sTREM2-depleted CM, or control medium for 24 h.

HEK293-tau cells were transfected with 50 nM siRNA using Lipofectamine 2000 according to the manufacturer's instructions. After 48 h, the cells were collected to assess the efficiency of RohA, ROCK and TG2 knockdown via qPCR and Western blotting. Cells were treated with Fc-sTREM2 (20 nM) or Fc control for 24 h after RhoA or ROCK siRNA or siNC treatment. The sequences of the siRNAs were RhoA (human)-siRNA: 5′-TGGAAAGACATGCTTGCTCAT-3′, ROCK (human)-siRNA: 5′-ATCAGAGGTCTACAGATGA-3′, and TG2 (human)-siRNA: 5′-GCATTAACACCACTGACAT-3′. TG2 (mouse)-siRNA: 5′-GCATTAACA CCACGGACAT-3′.

To test the role of the Rho-ROCK pathway in tau phosphorylation, HEK293-tau cells were treated with the control (Fc, 20 nM), sTREM2 (20 nM), RhoA inhibitor Tat-C3 (0.1 μg/ml, Cytoskeleton, CT03) or the ROCK inhibitor Y-27632 (25 nM, Santa Cruz, sc-3536) for 24 h. RhoA activator (1 U/ml, Cytoskeleton, CN01) was added to HEK293-tau cells after the cells were treated with Fc-sTREM2 (20 nM) for 24 h and incubated for another 30 min before analysis. For Western blot analysis, the cells were lysed with NP-40 Lysis Buffer (Beyotime) supplemented with a protease inhibitor cocktail on ice for 30 min. Lysates were centrifuged at 16,000 × g for 20 min at 4 °C. The concentrations of the supernatant were measured by BCA Protein Assay Kit (Pierce).

## Primary neuronal culture and treatment

Eighteen-day gestational wild-type mice or tau P301S mice were sacrificed by cervical dislocation. The embryos were removed quickly. The meninges and blood vessels were removed, and the cerebral cortex was isolated. The brain tissue was subsequently cut into pieces and resuspended in 3 mL of horse plus bovine serum medium (6% horse serum, 6% fetal bovine serum, 1% penicillin–streptomycin, 1% glutamine in Dulbecco's Modified Eagle Medium). The cell suspension was centrifuged at 100 × g at 4 °C for 5 min. The cells were resuspended in 5 ml of horse plus bovine serum medium and seeded into polylysine-coated six-well plates. After culturing for 4 h, the culture medium was changed to neuronal medium (2% B27 Supplement, 1% Penicillin-Streptomycin, 1% Glutamine in Neurobasal Media). After two days of culture, glial cells were removed by exposure to 10 μmol/L cytarabine for 6 h.

To investigate the effect of sTREM2 on tau phosphorylation, primary neurons from tau P301S mice were cultured for 14 days in vitro (DIV) and then exposed to Fc or Fc-sTREM2 (4 nM, 20 nM, 40 nM) for 24 h. To knockdown TG2, primary neurons from tau P301S mice were infected with rAAV-U6-shRNA(TAGLN2)-CMV-EGFP-SV40 poly40 or rAAV-U6-shNC-CMV-EGFP- SV40 poly40 at 5 DIV. The cells were then treated with Fc or Fc-sTREM2 (20 nM) for 24 h at 14 DIV. To detect the binding of sTREM2 and TG2 in neurons, TG2 knockdown was performed by transfecting TG2-siRNA or control siRNA at 12 DIV. The neurons were then incubated with Fc, Fc-sTREM2, FITC-control peptide, or FITC-peptide 1 for 30 min before detection.

## Reagents

Anti-p-tau S202 (ab108387, 1:1000), p-tau S396 (ab109390, 1:1000), p-GSK3β Y216 (ab75745, 1:1000), GSK3β (ab32391, 1:1000), and p-RhoA S188 (ab41435, 1:500) were purchased from Abcam (Cambridge, UK). Anti-GST (10,000-0-AP, 1:5000), GFP (66002-2-Ig, 1:5000), His (66005-1-Ig, 1:5000), RhoA (10749-1-AP, 1:500), GAPDH (60004-1-Ig, 1:10,000), and TG-2 (60044-1-Ig, 1:500), TG-2 (10234-2-AP, 1:500), anti-human IgG (16402-1-AP, 1:200 for immunofluorescence, 1:1000 for western blot) were purchased from Proteintech. The anti-TG-2 (sc-166697, 1:200)

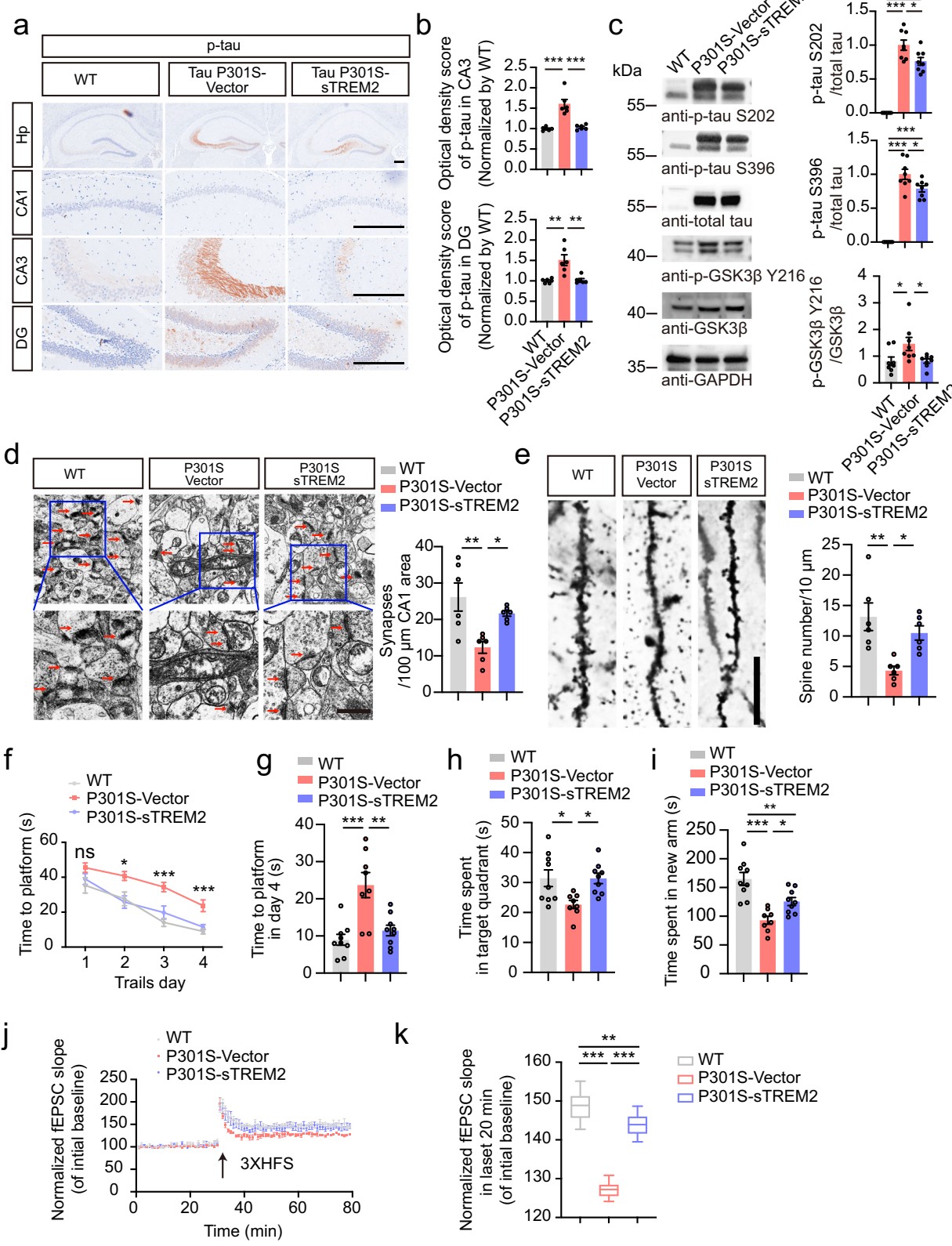

antibody was from Santa Cruz. Tau5 antibody (AHB0042, 1:1000), anti-p-Tau (Ser202, Thr205) (MN1020, 1:1000), anti-p-tau T181 (MN1050, 1:1000), Goat anti-Mouse IgG (H + L) Highly Cross-Adsorbed Secondary Antibody, Alexa Fluor 488 (A-11001, 1:1000), Goat anti-Rabbit IgG (H + L) Highly Cross-Adsorbed Secondary Antibody, Alexa Fluor 488 (A-11034, 1:1000), Goat anti-Rabbit IgG (H + L) Cross-Adsorbed Secondary Antibody, Alexa Fluor 594 (A-11012, 1:1000), Goat anti-Mouse

IgG (H + L) Cross-Adsorbed Secondary Antibody, Alexa Fluor 594 (A-11005, 1:1000) were purchased from ThermoFisher. Anti p-tau S404 (310196, 1:1000), and p-tau T231 (381181, 1:1000) were from Zenbio. Cell-permeable Rho inhibitor (C3 Trans based) (CT04-A), RhoA activator (Cytoskeleton, CN01), and Rhotekin RBD protein on GST beads (RT02-A) were purchased from Cytoskeleton. Y-27632 (sc-3536) was purchased from Santa Cruz Biotechnology. Protein A MagBeads

**Fig. 5 | sTREM2 attenuates tau pathology and memory loss in tau P301S mice.** p-Tau S202/T205 immunostaining (**a**) and quantification (**b**) of 7-month-old wild-type mice and tau P301S mice overexpressing AAV-vector or AAV-sTREM2. Scale bar, 250 μm. (mean ± s.e.m.; one-way ANOVA, $n = 6$ mice per group). **c** Western blots showing the phosphorylation of tau and GSK3β (mean ± s.e.m.; one-way ANOVA, $n = 8$ mice per group). **d** Electron microscopy of synapses in the hippocampus (left). Stars indicate synapses. Scale bar, 1 μm (mean ± s.e.m.; one-way ANOVA, $n = 6$ mice per group). **e** Golgi staining revealed the dendritic spines from the apical dendritic layer of the CA1 region (left). Scale bar, 15 μm. Quantification of spine density (right) (mean ± s.e.m.; one-way ANOVA, $n = 6$ mice per group). Morris water maze analysis of the escape latency (**f**) and escape latency on day 4 (**g**) (mean ± s.e.m.; $n = 9$ mice in WT, P301S-sTREM2 groups, $n = 8$ mice in P301S-Vector group; one-way ANOVA). **h** Probe trial of the Morris water maze test (mean ± s.e.m.; $n = 8$–9 mice per group; *$P < 0.05$, one-way ANOVA). **i** Y-maze analysis as time spent in new arms (mean ± s.e.m.; $n = 9$ mice in WT, P301S-sTREM2 groups, $n = 8$ mice in P301S-Vector group; one-way ANOVA). **j** The slope of fEPSPs after HFS recorded in hippocampal slices. The arrow indicates the onset of HFS. **k** Quantitative analyses of normalized fEPSPs at 60–80 min. Box–whisker plot (displaying the Max/Min at the whiskers, the 75/25 percentiles at the boxes, and the median in the center line). (mean ± s.e.m.; $n = 4$ mice per group; two-way ANOVA). *$P < 0.05$, **$P < 0.01$, ***$P < 0.001$. AAVs adeno-associated virus, HFS high-frequency stimulation, LTP long-term potentiation.

(L00464) were purchased from GenScript. TREM2 ELISA kits (SEK11084) were purchased from Sino Biological.

### Generation of the sTREM2 antibody

The sTREM2-specific antibody was generated by immunizing rabbits with a peptide near the C-terminus of sTREM2 (Ac-CGESESFEDAHVEH-OH). The rabbits were boosted 4 times with the immunizing peptides at 3-week intervals. The titers against the immunizing peptide were determined by ELISA. The antiserum was affinity purified using the immunogen peptide and counter-screened with full-length TREM2. The immunoactivity of the antibody was confirmed by WB and immunostaining.

### Expression constructs and AAV production

The following plasmids were used in this study: pFUSE-hIgG1e1-Fc2-sTREM2, GST-sTREM2 (1–158), GST-TREM2 (20–46), GST-TREM2 (47–76), GST-TREM2 (77–105), GST-TREM2 (106–136), pcDNA3.1-His-TG2, GST-RhoA, and GST-RhoA S188A.

AAV2/8 expressing human EGFP-2A-sTREM2-3×FLAG TREM2(1–171 aa) (AAV-sTREM2) or EGFP (control AAV) under the control of the CAG promoter was prepared by Obio Technology (Shanghai, China). The virus has been used previously to overexpress sTREM2 in the brain[44]. rAAV-U6-shRNA(TAGLN2)-CMV-EGFP-SV40 poly40 and rAAV-U6-shNC-CMV-EGFP-SV40 poly40 were purchased from Shumi Technologies (Wuhan, China).

### Purification of Fc-sTREM2 and Fc

The Fc-tagged human sTREM2 and Fc-vector were purified from the conditioned media of HEK293 cells as described previously[46]. Briefly, HEK293 cells were transfected with Fc-sTREM2 or Fc-vector. Forty-eight hours later, the conditioned media was incubated with Protein A resin (Genscript). The beads were washed with washing buffer (20 mM $Na_2HPO_4$, 0.15 M NaCl, pH 7.0) and eluted with elution buffer (0.1 M glycine, pH 2-3). The eluted proteins were dialyzed against PBS and filtered via a 0.22 μm filter before use. The protein concentrations were determined using a BCA kit (Pierce). To obtain heat-inactivated sTREM2, purified Fc-sTREM2 was boiled at 100 °C for 1.5 h.

### Affinity purification-mass spectrometry (AP-MS)

To identify the receptor of sTREM2, the membrane protein of SH-SY5Y cells was extracted using the Membrane Protein Extraction Kit (Beyotime Biotechnology, P0033) and incubated with Fc-sTREM2 and Protein A agarose for 24 h. To confirm the binding between sTREM2 and TG2 in neurons, the cell membrane fraction of primary neurons from WT mice or P301S mice was extracted and incubated with purified His-sTREM2 and Ni beads for 24 h. After extensive washing, the beads were added to the 50 μL reaction solution (1% SDC/100 mM Tris-HCl pH 8.5/10 mM TCEP/40 mM CAA) and boiled at 95 °C for 10 minutes. Fifty microliters of reaction solution, elution buffer, 100 μL of deionized water, and 1 μg of trypsin were added to the reaction and digested at 37 °C overnight. An equal volume of 1% formic acid/ethyl acetate was added to terminate the reaction, and SDB cartridges were used for desalting. Finally, 1.25% $NH_3H_2O$/80% ACN was used to elute the peptide fragments.

Mass spectrometry was performed using a TripleTOF 5600 LC/MS system from SCIEX. Samples were aspirated through an autosampler and bound to a C18 capture column (5 μm, 5 × 0.3 mm) and then eluted to an analytical column (75 μm × 150 mm, 3 μm particle size, 100 Å pore size, Eksigent) for separation. An analytical gradient (0 min in 5% B, 65 min of 5-23% B) was established using two mobile phases (mobile phase A: $H_2O$, 0.1% formic acid and mobile phase B: ACN, 0.1% formic acid), 20 min of 23–52% B, 1 min of 52–80% B, 80% B for 4 min, 0.1 min of 80–5% B, 5% B for 9.9 min. The flow rate of the liquid phase was set to 300 nL/min. For MS IDA mode analysis, each scan cycle consisted of a full MS scan (m/z range 350–1500, ion accumulation time 250 ms) followed by 40 MS/MS scans (m/z range 100 ms) −1500, ion accumulation time 50 ms. The MS/MS acquisition conditions were set as the precursor ion signal greater than 120 cps and the charge number from +2 to +5. The exclusion time for repeated ion acquisition was set to 18 s. Mass spectral data generated by TripleTOF 5600 were searched by ProteinPilot (V4.5) using the database search algorithm Paragon. The database used for the search was the *Homo sapiens* or *Mus musculus* Proteome Reference Database in UniProt.

### Western blot analysis

Cells were lysed in nonidet-P40 (NP40) lysis buffer (Beyotime) supplemented with protease and phosphatase inhibitors. The protein concentrations were determined using the BCA protein assay kit (Thermo Fisher). The cell lysates were boiled in SDS loading buffer. The samples were subjected to SDS-PAGE and transferred to a nitrocellulose membrane. The membranes were incubated with primary antibodies overnight at 4 °C. After being washed in TBST and incubated with horseradish peroxidase (HRP)-conjugated secondary antibodies, the membranes were visualized using enhanced chemiluminescent (ECL) substrates.

### GST pull-down assay

The cell lysates overexpressing GST-sTREM2 or GST-tagged sTREM2 fragments were incubated with glutathione agarose for 4 h at room temperature. After washing 4 times with NP-40, the beads were incubated with cell lysates overexpressing His-tagged TG2 overnight at 4 °C. The beads were washed 4 times with NP-40, boiled in SDS loading buffer, and analyzed by immunoblotting.

### Cell-surface binding assays

HEK293 cells were transiently transfected with EGFP-vector or EGFP-TG2. Twenty-four hours later, the cells were incubated with purified Fc-tagged sTREM2 and anti-human Fc antibodies at a final concentration of 7.5 ng/ml at room temperature for 30 min. After washing 3 times with PBS, the coverslips were fixed and processed following regular immunostaining procedures[47,48].

### Immunostaining

Brain tissues were fixed with 4% paraformaldehyde and embedded in paraffin. The sections were deparaffinized followed by antigen retrieval. For the immunostaining of cells, the slides were directly fixed with 4% paraformaldehyde. Then, the brain sections or cell slides were blocked

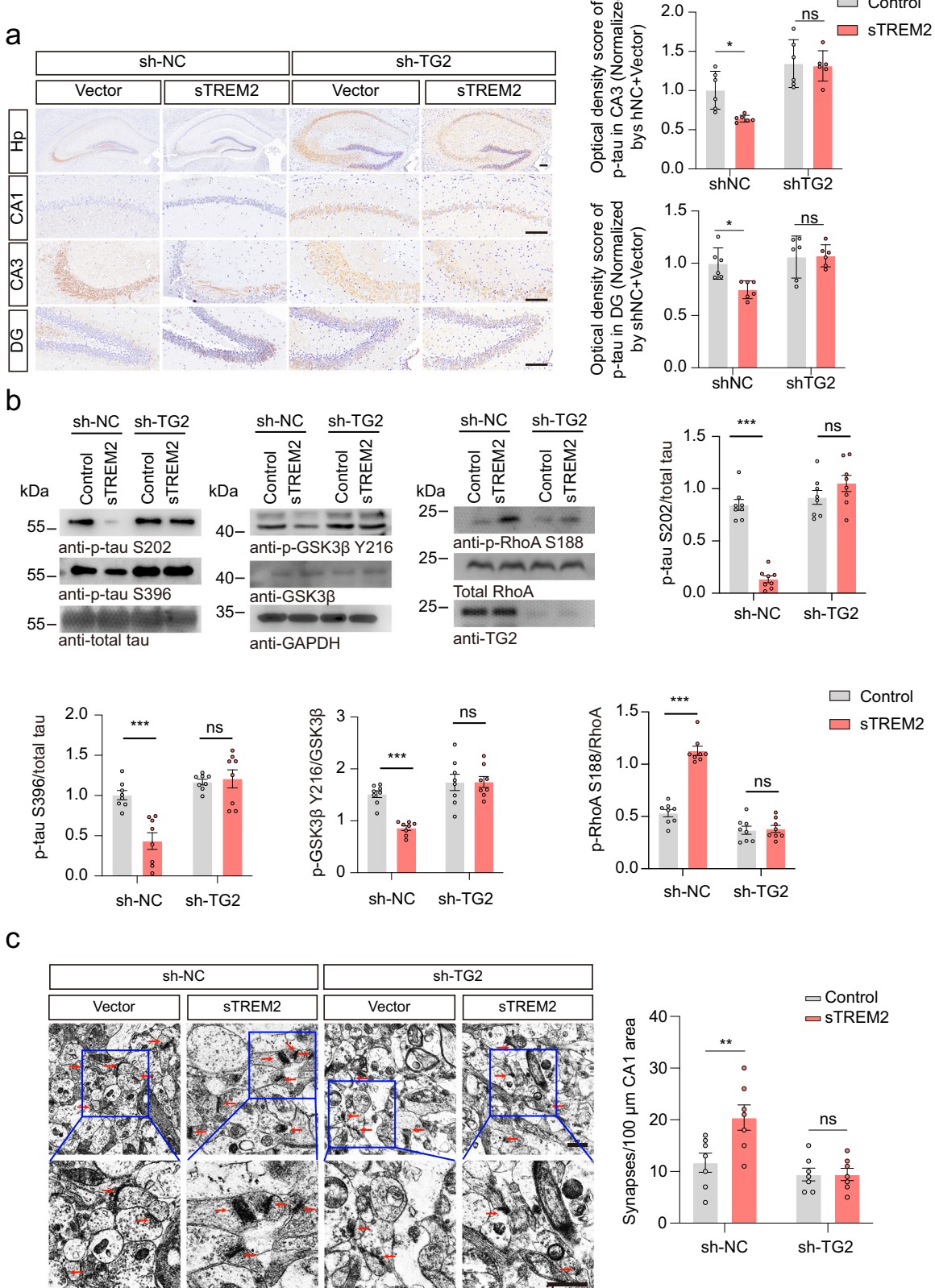

**Fig. 6 | Knockdown of TG2 abolishes the inhibitory effect of sTREM2 on tau phosphorylation. a** p-Tau S202/T205 immunostaining and quantification of 7-month-old mice injected with sh-TG2 and sTREM2. Scale bar, 250 μm (mean ± s.e.m.; two-way ANOVA, $n = 6$ mice per group). **b** Western blots showing the phosphorylation of tau, GSK3β, and RhoA (mean ± s.e.m.; two-way ANOVA, $n = 8$ mice per group). **c** Electron microscopy of synapses in the hippocampus (left). Stars indicate synapses. Scale bar, 1 μm. Quantification of synaptic density (right) (mean ± s.e.m.; two-way ANOVA, $n = 6$ mice per group). *$P < 0.05$, **$P < 0.01$, ***$P < 0.001$.

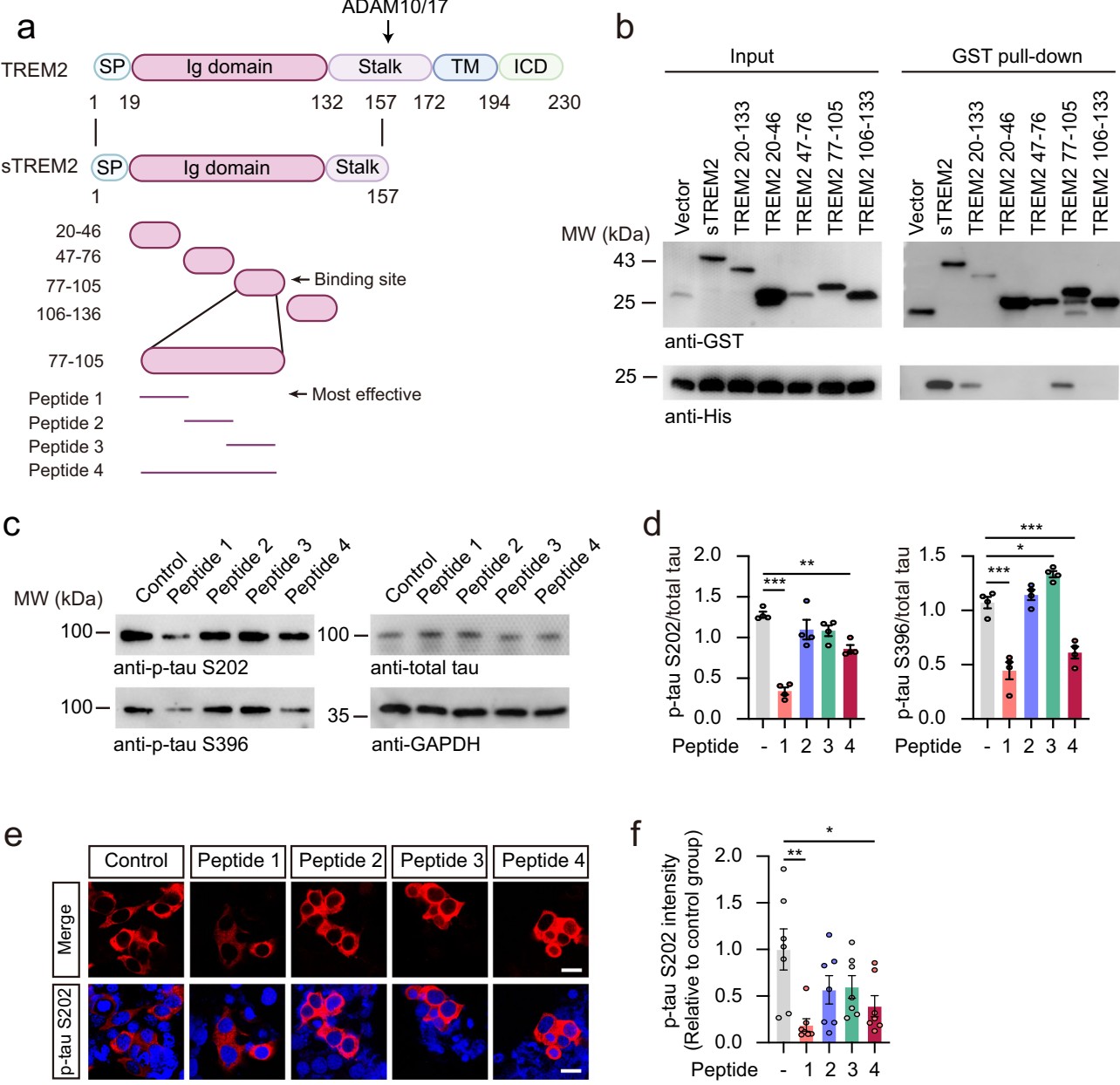

**Fig. 7 | A short peptide of sTREM2 suppresses tau hyperphosphorylation in vitro. a** Schematic illustration of sTREM2 fragments. **b** GST pull-down assay showing the interaction between GST-sTREM2 fragments and His-TG2. $n = 4$ independent experiments. **c**, **d** Western blots showing the phosphorylation of tau and GSK3β in HEK293-Tau cells treated with different sTREM2 peptides (10 μM) for 24 h (mean ± s.e.m.; one-way ANOVA, $n = 4$ independent experiments; compared with the control group). **e** p-Tau S202 immunostaining of HEK293-Tau cells treated with different sTREM2 peptides (10 μM) for 24 h. Scale bars, 15 μm. **f** Quantification of p-tau S202 (mean ± s.e.m.; one-way ANOVA, $n = 7$ independent experiments; compared with the control group). *$P < 0.05$, **$P < 0.01$, ***$P < 0.001$.

and permeabilized in PBS with 5% BSA and 0.3% Triton X-100 for 30 min and incubated with primary antibodies overnight at 4 °C. The signal was developed using a Histostain-SP kit for immunohistochemistry (Invitrogen). For immunofluorescence, the sections or cell slides were incubated with Alexa Fluor 594-conjugated anti-rabbit or Alexa Fluor 488-conjugated anti-mouse antibodies (Abcam). The nuclei were stained using 4′,6-diamidino-2-phenylindole (DAPI). A Multiplex fluorescence immunohistochemical staining kit (Absin, Catalog No.abs50028) was used for multiple fluorescence immunostaining.

The immunofluorescence images were quantified using ImageJ. The integrated pixel intensity of the region of interest (ROI) was quantified, and the background fluorescence intensity was subtracted. IHC images were quantified through ImageJ and the IHC Profiler plugin. The program counts the pixels and evaluates the percentage

contributions of high positive, positive, low positive, and negative areas. The 'optical density score' was calculated as (percentage contribution of high positive*4 + percentage contribution of positive*3 + percentage contribution of low positive*2 + percentage contribution of negative*1)/100.

## GTP-RhoA pull-down assay

Cells were cultured in serum-free medium for 12 h and then incubated with the indicated concentrations of sTREM2 for 24 h. The levels of total RhoA were analyzed by western blotting. For the GTP-RhoA pull-down assay, the cell lysates were incubated with Rhotekin RBD protein on GST beads for 3 h at 4 °C. The beads were washed 4 times and separated by SDS-PAGE to determine the levels of active RhoA. The activity of RhoA was quantified by the ratio of active RhoA divided by total RhoA.

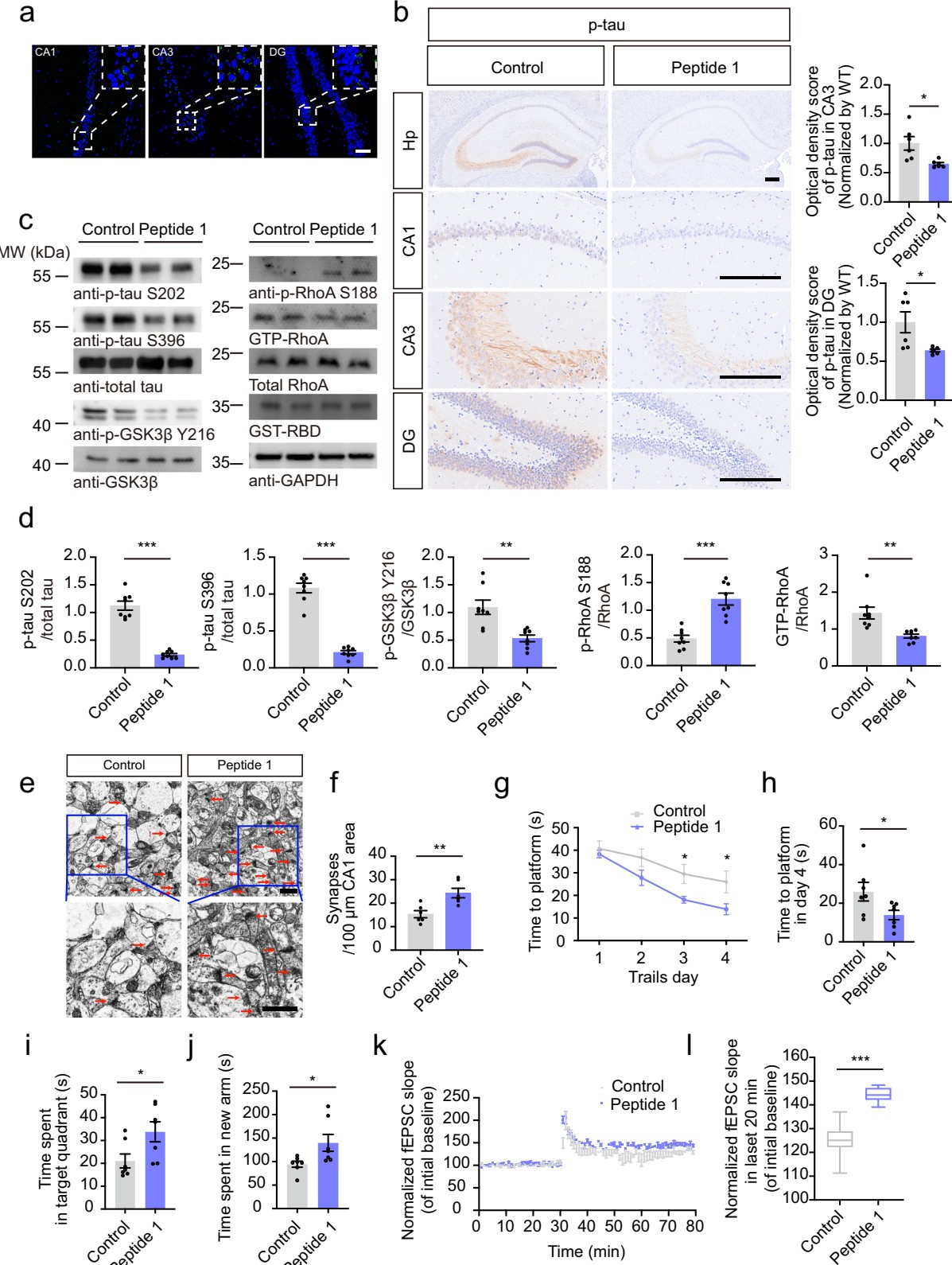

### Stereotaxic injection

Three-month-old tau P301S mice were anesthetized and stereo-tactically injected with 200 nl of virus suspension containing $1 \times 10^9$ vector genomes (vg) into the bilateral hippocampus at coordinates posterior 2.5 mm, lateral 2.0 mm, and ventral 1.7 mm relative to bregma. The needle remained in place for 5 min before it was removed

slowly (throughout 2 min). The mice were placed on a heating pad until they recovered from the surgery.

### Administration of Tat-sTREM2 (77–89)

The peptides, including sTREM2 (77–89), sTREM2 (89–77), Tat-sTREM2 (77–89), Tat-sTREM2 (89–77), Tat-sTREM2 (77–89)-FITC,

**Fig. 8 | The active sTREM2 peptide suppresses tau pathology in tau P301S mice.** **a** FITC fluorescence in the hippocampus of tau P301S mice one week after i.p. injection of FITC-labeled sTREM2 (77-89) (Peptide 1). Scale bar, 50 μm. $n = 6$ mice per group. **b** p-Tau S202/T205 immunostaining and quantification of tau P301S mice administrated with sTREM2 (89-77) or (77-89). Scale bar, 250 μm (mean ± s.e.m.; $n = 6$ mice per group; two-tailed t-test). **c, d** Western blots and quantification showing the phosphorylation of tau and GSK3β (mean ± s.e.m.; $n = 8$ mice per group; Two-tailed t-test). **e** Electron microscopy of synapses in the hippocampus. Stars indicate synapses. Scale bar, 1 μm. **f** Quantification of synaptic density (mean ± s.e.m.; Two-tailed t-test; $n = 6$ mice per group). Morris water maze analysis of the escape latency (**g**) and escape latency on day 4 (**h**) (mean ± s.e.m.; $n = 7$ mice per group; Two-tailed t-test). **i** Probe trial of Morris water maze test (mean ± s.e.m.; $n = 7$ mice per group; Two-tailed t-test). **j** Y-maze analysis as time spent in new arms (mean ± s.e.m.; $n = 7$ mice per group; Two-tailed t-test). **k** The slope of fEPSPs after HFS recorded in hippocampal slices. The arrow indicates the onset of HFS. **l** Quantitative analyses of normalized fEPSPs at 60–80 min. Box–whisker plot (displaying the Max/Min at the whiskers, the 75/25 percentiles at the boxes, and the median in the center line). (mean ± s.e.m.; $n = 4$ mice per group; Two-tailed t-test). $*P < 0.05$, $**P < 0.01$, $***P < 0.001$. AAVs adeno-associated virus, HFS high-frequency stimulation, LTP long-term potentiation.

and Tat-sTREM2 (89–77)-FITC, were synthesized by ChinaPeptides (Shanghai, China). The purity of the peptides was higher than 99%. Three-month-old tau P301S mice were i.p. administered 10 mg/kg Tat-sTREM2 (77-89) or Tat-sTREM2 (89-77) once every 5 d for 4 months.

## Electron microscopy of synapses
Electron microscopy was used to detect the synaptic density as previously described[49]. Mice were anesthetized and perfused with 2% glutaraldehyde. Then, the hippocampal slices were postfixed in cold 1% $OsO_4$ for 1 h and prepared using standard procedures. Ultrathin sections (90 nm) were stained with uranyl acetate and lead acetate and viewed at 100 kV in a JEOL 200CX electron microscope. Synapses were identified by the presence of synaptic vesicles and postsynaptic densities.

## Golgi staining
After perfusion and fixation in 10% formalin, the mouse brains were postfixed in 10% formalin for 24 h, immersed in 3% potassium bichromate at room temperature for 7 d in the dark, and transferred into 2% silver nitrate solution for 3 d in the dark. The hippocampus was cut into 80–200 μm sections with a cryostat (Leica, Germany). The sections were mounted on slides, dried at room temperature, and dehydrated by washing with 70%, 90%, 100% ethanol, and 100% xylene for 6 min each. Images were captured using a microscope.

## Electrophysiology
The brain slices of mice were prepared as previously described[49]. Briefly, mouse hippocampi were dissected and cut into 350 μm sections. For LTP recording, the electrophysiological signals were acquired using the MED64 System (Alpha MED Science, Panasonic). The field excitatory postsynaptic potentials (fEPSPs) in CA1 neurons were recorded by stimulating the Schaeffer fibers from CA3. LTP was induced by applying 3 trains of high-frequency stimulation (HFS; 100 Hz, 1 s duration).

## Morris water maze test
The mice were trained in a round water-filled tub with extra maze cues as described previously[50]. Each subject was tested 4 trials/day for 5 consecutive days with a 15 min intertrial interval. If the subjects did not reach the platform within 60 s, they were manually guided to it. After 4 d of task acquisition, a probe trial was performed on day 5. During the probe trial, the platform was removed, and the percentage of time spent in the quadrant was measured over 60 s. All trials were analyzed for latency and swim speed by ANY-Maze software (San Diego Instruments).

## Y-maze test
The start arm (the mice started to explore, always open), novel arm (blocked during the first trial but open during the second trial), and another arm were randomly designated. In the first trial (training), the novel arm was blocked, and the mice were allowed to explore the start arm and another arm for 5 min. After a 2-h intertrial interval, the second trial (retention) was conducted, and the mice were free to access the three arms for 5 min. The number of entries and time spent in each arm were analyzed by ANY-Maze software.

## Statistical analyses
Data were expressed as means ± SEM. Statistical analysis was performed using either Two-tailed t-test (two-group comparison), one-way ANOVA followed by LSD post hoc test (more than two groups), or two-way ANOVA (two categorical independent variables). All statistical tests were two-sided. Differences with $P$ values less than 0.05 were considered significant.

## Reporting summary
Further information on research design is available in the Nature Portfolio Reporting Summary linked to this article.

## Data availability
All data in this study are included in the manuscript and supporting files. The *Mus musculus* Proteome Reference Database was from UP000000589, while the *Homo sapiens* Proteome Reference Database was from UP000005640. Source data are provided with this paper.

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

## Acknowledgements

This work was supported by the National Key Research and Development Program of China (No. 2019YFE0115900, to Z.Z.), the National Natural Science Foundation of China (No. 82271447 and 81822016, to Z.Z., 82301356 to X.Z.), the Innovative Research Groups of Hubei Province (2022CFA026, to Z.Z.), the "New 20 Terms of Universities in Jinan" grant (No. 202228022, to Z.Z.), China Postdoctoral Science Foundation (2021M702523 to X.Z.), Knowledge Innovation Program of Wuhan-Shuguang Project (2022020801020481 to X.Z.). We thank Dr. Xiao-Xin Yan (Central South University Xiangya School of Medicine) for preparing the postmortem brain samples.

## Author contributions

Project design: Z.Z., X.Z. Investigation: X.Z., L.T., J.Y., L.M., J.C., L.Z., J.W., M.X. Data analysis: X.Z., Z.Z. Funding acquisition: Z.Z., X.Z. Project administration: Z.Z. Writing – original draft: X.Z. Writing – review & editing: Z.Z.

## Competing interests

The authors declare no competing interests.
