## [Peer Review File · Nature Communications]

Reviewers' comments:

Reviewer #1 (Remarks to the Author):

The authors reported a novel function of soluble TREM2 (sTREM2) in ameliorating tau phosphorylation in neurons and alleviating cognitive deficits in tauopathy mouse models. They showed that the action of sTREM2 is through binding to TAGLN2 receptor and RhoA-ROCK-GSK3b signaling. The study has mapped the activity of the sTREM2 to a short peptide (77-89) and demonstrated the protective function of such peptide when administered into the tauopathy model. The current observation may provide insight into the mechanism whereby sTrem2 offers neuroprotection. Despite the interesting observations, the current version of the manuscript can be improved by addressing the following concerns. In addition, important information for critical reagents, postmortem brains, and detailed procedure are missing.

1. The knowledge of neuronal function of TG2 (TAGLN2) is extremely limited. According to the CNS cells gene expression profile (<https://www.brainrnaseq.org/>), TAGLN2 is expressed primarily in endothelial cells, microglia, and OPC, while its expression is low in neurons. Thus, it is surprising to see the strong effect of TG2 in mediating sTREM2 action in neurons. Can the authors supply evidence that deletion of neuronal TG2 abolishes the binding of sTREM2 or peptide 77-89 to the neurons? are there additional receptors that could also bind sTREM2 in neurons?

2. TG2 was reported as a cytoplasmic small actin-binding protein. The evidence for its localization at neuron surface as a receptor is not strong based on the current study. Additional study may be needed to support its role as a receptor.

3. Can the authors provide information about the distribution of FITC-conjugated peptide 1 in the brain after intraperitoneal injection? The results are derived from the hippocampus. Fig. 8A does not contain any of that information.

4. Does KD of TG2 affect p-tau under basal condition (Figure 3E-H)?

5. The authors should provide quantification of p-tau for vivo images in Fig. 5A, Fig. 6A, and Fig. 8B.

6. The quality of TG2 blot in postmortem tissues should be improved (Fig. S7A). It is not equally developed.

7. Questions about control experiments:

1) The authors did not include any control peptide or proteins when they examined the effect of sTrem2-Fc fusion protein on p-tau levels in Fig. 1.

2) Lack of information for the "control" (ex. Fig. 3E-H, Fig. 4). The authors had limited information about sTrem2-Fc fusion proteins in the method section and no information is provided for the control peptides.

3) The authors should provide evidence of the increased levels of sTrem2 in the brain after AAV-sTREM2 injection. Also, it would be great to visualize the infected cells or area for both AAV-sTREM2 and AAV-sNC.

4) The authors depleted sTREM2 from the medium of BV2-overexpression sTREM2. It is required to show the levels of sTREM2 before and after depletion.

8. lack of information in the materials and methods.

1) The author depleted the cleaved sTREM2 from the medium of BV2-overexpressing sTrem2. The antibody used for depletion and the strategy to overexpress sTREM2 in BV2 (vector information and transfection method for BV2, and how to deplete) are all missing.

2) The study stained sTREM2 in vivo (Fig. 2C, D) without sharing the information for sTREM2-specific antibody (should be validated).

3) No antibody information for detecting Fc protein (Fig. 2E).

4) No detailed information on virus production (promoter, virus serotype, and so on).

5) No detail on how they quantify the images in vitro and in vivo.

6) No information for the AD human brains and the brain areas that were used in the study.

Reviewer #2 (Remarks to the Author):

In the present study the authors aim to demonstrate that sTREM2 interacts with transgelin-2 (TG2) in neurons. sTREM2 binding to TG2 resulted in TG2 activation and inhibition of the RhoA-ROCK-GSK3 β pathway, leading to reduced tau hyperphosphorylation in neurons. The authors also identified the minimal active sequence of sTREM2 responsible for the protective effect on tau phosphorylation (sTREM2 (77-89)). Overall, this is an interesting manuscript describing a novel potential sTREM2 interactor and a protective pathway in the context of tau pathology. While different in vitro and in vivo approaches were used to test the role of sTREM2 on tau phosphorylation, the work has some major weakness, such as the lack of critical experiments proving the specificity of sTREM2 and TG2 interaction in neurons, and the incomplete characterisation of TG2 expression/ TG2-sTREM2 interaction in other brain cells.

Below, specific points of critique:

Major comments:

- The Authors state that “silencing of brain TREM2 exacerbates tau pathology^{15,16,17}, while overexpression of TREM2 ameliorates tau pathology¹”. This sentence is over-simplistic and does not consider the literature reporting a protective effect of TREM2 deficiency in models of tauopathy or showing a differential effect played by TREM2 in the context of β -amyloid and tau pathologies (PMID: 36219197; PMID: 29073081; PMID: 33675684; PMID: 30283031).

- Figure 1: Since different phosphorylation residues have been observed in Tauopathies (Wegmann et al., 2021 PMID: 33892381), authors should explain in the main text and discussion why they focused on S202 and S396 residues. Did the authors investigate whether sTREM2 treatment decrease tau phosphorylation also at other residues?

-Figure 1A: The authors generated a HEK293 cell line stably overexpressing GFP-Tau. They should provide immunofluorescence (IF) staining of GFP-Tau signal and show co-localization between GFP-Tau and p-S202 and p-S396 residues.

-Figure 1A-E: Is the control group (sTREM2= 0 nM) treated with heat-inactivated sTREM2? The use of heat-inactivated sTREM2 has been previously described in PMID: 28209725. To have the proper control, the authors should use heat-inactivated sTREM2 or at least add the same vehicle in which sTREM2 is purified.

-Fig. 1: Please specify in the M&M what sTREM2 has been used.

How do the authors prove that sTREM2 is added at increasing concentrations to HEK293 cell line and to neuronal cultures? They should take advantage of a tagged- sTREM2 (e.g., HIS-tagged as used in Fig. 2) or measure sTREM2 levels by ELISA.

-Fig. 1D: Does sTREM2 addition to the media affect neuronal health? The authors should add a neuronal marker (e.g. MAP2, NeuN) and stain for propidium iodide/ Calcein to verify whether sTREM2 is affecting neuronal survival. The authors should also perform the same experiment using WT neurons (at least testing one dosage of sTREM2).

-Fig. S1: There is no mention in the M&M of which anti-TREM2 antibody has been used to deplete the CM of sTREM2.

-Figure 2: Why do the authors use SH-SY5Y cells instead of primary neurons from WT mice? WT primary neurons are more suitable to pull down neuronal cell membrane interactors and should be preferred.

Also, Mass Spect on primary neurons from tau P301S transgenic mice would help clarifying whether sTREM2 and TG2 interaction is modulated during AD. If the authors cannot address this second point, they should at least discuss it in the manuscript.

- Fig. 2: The authors should verify the purification of Fc-tagged sTREM2 by western blot and provide the purity of cell membrane fraction (enrichment in membrane proteins and absence of nuclear proteins).

-Table S1: LC MS/MS analysis of sTREM2 interactors found using the cell membrane fraction extracted from SH-SY5Y cells revealed that TREM2 protein interacts with sTREM2 with the highest affinity ($\text{Log}_2(\text{FcsTREM2}/\text{Fc}) = 10.4757$). How do the authors explain the expression of TREM2 protein in SH-SY5Y neuronal cells? This is a particularly crucial point since no one has previously reported TREM2 protein expression in neurons.

-Fig. 2C and D: What brain area is represented in the images? The authors should add lower magnification images of the brain area analysed. Also, adding a neuronal marker other than NeuN (e.g. Map2) would help clarifying where sTREM2 and TG2 localize on neurons since NeuN is a DNA-binding protein and it is not expressed at the neuronal membrane. In this regard, could the authors show whether sTREM2 and TG2 are present on the cell soma only (as it appears from the confocal images in Fig 2C and D) or are expressed on axons/ branches too?

The signal of TG2 in Fig. 2D cannot be clearly distinguished from the background. Confocal images showing TG2 expression in mouse brains with a better resolution and lower background should be provided. Also, 3D reconstruction of confocal z-stack would help strengthening the authors' finding.

- TG2 is mostly expressed by smooth muscle cells and in the brain, RNA-seq datasets show that TAGLN2 gene is expressed at high levels by microglia/endothelial cells rather than neurons (<http://www.brainrnaseq.org/>). The authors should show if TG2-sTREM2 interaction occurs in glial cells by using microglial (IBA1, P2Y12 receptor) and astrocytic (GFAP) markers. They should also discuss the meaning of this interaction in both glial cells and neurons in the context of tau pathology.

-Figure S7: Bands for TG2 protein and p-RhoA S188 by WB in AD patients are not very convincing. The authors should show clearer TG2 and p-RhoA S188 representative bands.

-S7: Could the authors perform immunofluorescence staining of TG2 together with a neuronal marker in AD versus control patients?

-Figure 3E: The authors should provide WB and/or IF staining to confirm TG2 silencing upon shRNA treatment.

-Fig. S8: The authors should add an image of hippocampus from mice injected with control AAVs in parallel to the AAV-EGFP-sTREM2 injected ones.

-Fig. 5 and 6: the authors should provide IF staining for inhibitory and excitatory synapses (e.g. PSD-95 and VGLUT-1) to test whether both compartments are involved.

Figure 5D: In the EM images, it is not clear what the red stars are pointing. Could the authors provide images with a better resolution? Since it looks like that the red stars cover the synaptic sites, replacing them with smaller stars/arrows would help the reader understanding the figure better.

Fig. S9: As previously pointed out, the authors should provide lower magnification images to have a better picture of TG2 expression in the hippocampal region. They should also add a neuronal marker to show TG2 localization on neurons.

Fig. 8A and S13B: Representative images of FITC fluorescence in the brain of tau P301S mice injected i.p. with Tat-sTREM2 (77-89) or Tat-sTREM2 (89-77) are shown. The authors should stain these sections with an anti-TG2 antibody and with a neuronal marker (e.g., NeuN, Map2). This experiment will help understanding whether the FITC-peptide localizes at specific sites of the cell and colocalizes/ binds TG2 as observed for sTREM2.

Minor comments:

- HEK293 and SH-SY5Y cell culture and treatment should be more detailed in the M&M section.
- Please provide the expanded definition for SH-SY5Y cells.
- Fig.1D –3D: the authors should provide more information about neuronal culture and treatment in the M&M section.
- Fig.1 – S4 – 7E: It should be clarified for how long the cells are incubated with sTREM2.
- Figure: S1, S2, S4, S5: Letters in the figure legends are missing.
- Figure 4B,4D,4F: please provide details about dosages, time of treatment and specify the name of RhoA activator.
- Figure 5: The authors should clarify in the main text or in the figure legend whether mice were analysed at 7 months of age for all the tests performed.
- Fig. S8: Please, specify in the fig. legend at what age tau P301S mice were injected with AAV-EGFP-sTREM2.
- Fig. S13: Please, specify in the fig. legend which brain area is represented.
- Fig. 8G and 5F: statistical significances are missing in the graphs.

Point-by-point response to the reviewers' comments

Reviewer #1:

The authors reported a novel function of soluble TREM2 (sTREM2) in ameliorating tau phosphorylation in neurons and alleviating cognitive deficits in tauopathy mouse models. They showed that the action of sTREM2 is through binding to TAGLN2 receptor and RhoA-ROCK-GSK3b signaling. The study has mapped the activity of the sTREM2 to a short peptide (77-89) and demonstrated the protective function of such peptide when administered into the tauopathy model. The current observation may provide insight into the mechanism whereby sTrem2 offers neuroprotection. Despite the interesting observations, the current version of the manuscript can be improved by addressing the following concerns. In addition, important information for critical reagents, postmortem brains, and detailed procedure are missing.

Response: We appreciate the reviewer's insightful and constructive comments and advice. We have carefully addressed the reviewer's concerns and properly revised the manuscript. These comments and suggestions enabled us to provide a highly improved manuscript.

1. The knowledge of neuronal function of TG2 (TAGLN2) is extremely limited. According to the CNS cells gene expression profile (<https://www.brainrnaseq.org/>), TAGLN2 is expressed primarily in endothelial cells, microglia, and OPC, while its expression is low in neurons. Thus, it is surprising to see the strong effect of TG2 in mediating sTREM2 action in neurons. Can the authors supply evidence that deletion of neuronal TG2 abolishes the binding of sTREM2 or peptide 77-89 to the neurons? are there additional receptors that could also bind sTREM2 in neurons?

Response: We appreciate the reviewer's comments. It is true that knowledge of the neuronal function of TG2 is very limited. Although the mRNA level of TG2 in neurons is relatively low as detected by RNAseq (<https://www.brainrnaseq.org/>), a high-resolution mass spectrometry-based proteomics for in-depth analysis of the major brain regions and cell types identified that TG2 is evenly expressed in neurons and glial cells (Nature Neuroscience, 2015, 18:1819-1831). We have added this reference to the manuscript (**Page 7, lines 277-280**). Furthermore, we detected the expression of TG2 in primary neurons through mass spectrometry (**Fig. 2B, C, Tables S2, S3**), Western blotting ("**input, anti-transgelin2**" in **Fig. 2I, J**), and immunostaining (**Fig. 2D-F**). As suggested by the reviewer, we transfected primary neurons with TG2 siRNA and found that downregulation of neuronal TG2 abolished the binding of sTREM2 and peptide 77-89 to neurons (**Fig. S4F**), indicating that TG2 is the major receptor for sTREM2.

Fig. 2

Supplementary Fig. 4F

Table S2

Accession	Gene	Intensity (control)	Intensity (sTREM2)	sTREM2/control	Log2 (sTREM2/control)
Q9WVA4	Tagln2	9.76	15.62	58.32	5.87

Table S3

Accession	Gene	Intensity (control)	Intensity (sTREM2)	sTREM2/control	Log2 (sTREM2/control)
Q9WVA4	Tagln2	8.71	15.47	108.03	6.76

2. TG2 was reported as a cytoplasmic small actin-binding protein. The evidence for its localization at neuron surface as a receptor is not strong based on the current study. Additional study may be needed to support its role as a receptor.

Response: TG2 has been reported to be expressed on the membrane of dendritic cells, HepaRG cells, T cells, and airway smooth muscle cells (**ref. 1-4**). In airway smooth muscle cells, TG2 has been identified as a receptor for extracellular ligands and induces RhoA phosphorylation (**ref. 4**). To further verify that TG2 is expressed on the surface of neurons, we separated the membrane fractions from primary neurons and perform WB to confirm its localization on the membrane (“**input, anti-transgelin2**” in **Fig. 2I**). In addition, we separated the membrane fractions from the primary neurons of WT mice and tau P301S transgenic mice, and performed affinity purification-mass spectrometry with purified sTREM2. We found that sTREM2 interacted with TG2 expressed on neuronal membranes (**Fig. 2C, Tables S2, S3**). As mentioned in our response to Comment #1, the downregulation of TG2 from primary neurons abolished the binding of sTREM2 to neurons (**Fig. S4F**). These experiments confirm that TG2 acts as the receptor for sTREM2.

References

1. Ferret-Bernard S, Castro-Borges W, Dowle AA, Sanin DE, Cook PC, Turner JD, MacDonald AS, Thomas JR, Mountford AP. Plasma membrane proteomes of differentially matured dendritic cells identified by LC-MS/MS combined with iTRAQ labelling. *Journal of Proteomics*, 2012, 75(3): 938–948.
2. Sokolowska I, Dorobantu C, Woods AG, Macovei A, Branza-Nichita N, Darie CC. Proteomic analysis of plasma membranes isolated from undifferentiated and differentiated HepaRG cells. *Proteome Science*, 2012, 10(1): 47.
3. Na BR, Kim HR, Piragyte I, Oh HM, Kwon MS, Akber U, Lee HS, Park DS, Song WK, Park ZY, Im SH, Rho MC, Hyun YM, Kim M, Jun CD. TAGLN2 regulates T cell activation by stabilizing the actin cytoskeleton at the immunological synapse. *The Journal of Cell Biology*, 2015, 209(1): 143–162.
4. Yin LM, Xu YD, Peng LL, Duan TT, Liu JY, Xu Z, et al. Transgelin-2 as a therapeutic target for asthmatic pulmonary resistance. *Science Translational Medicine*, 2015, 10: eaam8604.

3. Can the authors provide information about the distribution of FITC-conjugated peptide 1 in the brain after intraperitoneal injection? The results are derived from the hippocampus. Fig. 8A does not contain any of that information.

Response: In Fig. 8A, we have provided images of FITC fluorescence in the hippocampus of tau P301S mice one week after i.p. injection of FITC-labeled sTREM2 (77-89). As suggested by the reviewer, we fully characterized the distribution of peptide 1 in different brain areas after intraperitoneal injection (**Fig. 8A, Fig. S13C**).

Fig. 8A

Fig. S13C

4. Does KD of TG2 affect p-tau under basal condition (Figure 3E-H)?

Response: As suggested by the reviewer, we tested the effect of TG2 knockdown in wild-type neurons under basal conditions (Fig. S5E). Although tau phosphorylation levels were low in primary neurons from wild-type mice, sTREM2 slightly decreased tau phosphorylation. Knockdown of TG2 abolished the effect of sTREM2. In addition, the knockdown of TG2 directly increased tau phosphorylation.

Fig. S5E

5. The authors should provide quantification of p-tau for vivo images in Fig. 5A, Fig. 6A, and Fig. 8B.

Response: We appreciate the reviewer's comments. We quantified the levels of p-Tau in Fig. 5B, Fig. 6B, and Fig. 8C.

Fig. 5B

Fig. 6A

Fig. 8B

6. The quality of TG2 blot in postmortem tissues should be improved (Fig. S7A). It is not equally developed.

Response: We appreciate the reviewer's comments. We provided more representative Western blot images of TG2 in postmortem brain tissues (**new Fig. S8A**).

7. Questions about control experiments:

1) The authors did not include any control peptide or proteins when they examined the effect of sTrem2-Fc fusion protein on p-tau levels in Fig. 1.

Response: We appreciate the reviewer's comments. In Figure 1, we used Fc-tagged sTREM2 to treat HEK293-Tau cells and primary cortical neurons. The control cells were treated with Fc. We labeled this in Figure 1. Furthermore, we also treated the cells with heat-inactivated sTREM2 to show the specificity of sTREM2 on tau phosphorylation (Fig. 1A-D, Fig. S1C).

Fig. 1A-D

Fig. S1C

2) Lack of information for the “control” (ex. Fig. 3E-H, Fig. 4). The authors had limited information about sTrem2-Fc fusion proteins in the method section and no information is provided for the control peptides.

Response: We apologize for the confusion. In Fig. 3E-H and Fig. 4, the cells were treated with Fc-sTREM2 or Fc (control). We have added this information to the figure legends. The purification of the sTrem2-Fc fusion protein and Fc was described in the Methods section (**Page 10, lines 411-419**).

3) The authors should provide evidence of the increased levels of sTrem2 in the brain after AAV-sTREM2 injection. Also, it would be great to visualize the infected cells or area for both AAV-sTREM2 and AAV-sNC.

Response: As suggested by the reviewer, we showed the increased expression of sTREM2 in the brain after AAV-sTREM2 expression by Immunostaining and ELISA (**Fig. S9A, C**). The infected cells for AAV-sTREM2 and AAV-sNC are shown in **Fig. S9B**.

Fig. S9A

Fig. S9C

4) The authors depleted sTREM2 from the medium of BV2-overexpression sTREM2. It is required to show the levels of sTREM2 before and after depletion.

Response: We appreciate the reviewer’s comments. We quantified the levels of sTREM2 before and after depletion using ELISA (**Fig. S2C**).

Fig. S2C

8. lack of information in the materials and methods.

1) The author depleted the cleaved sTREM2 from the medium of BV2-overexpressing sTrem2. The antibody used for depletion and the strategy to overexpress sTREM2 in BV2 (vector information and transfection method for BV2, and how to deplete) are all missing.

Response: We used the TREM2 antibody (Proteintech, Ag26319) to deplete sTREM2 from the medium of BV2 cells overexpressing sTREM2. To overexpress sTREM2, BV2 cells were infected with AAV-sTREM2. We described the detailed methods in the revised manuscript (**Pages 8-9, lines 337-343**).

2) The study stained sTREM2 in vivo (Fig. 2C, D) without sharing the information for sTREM2-specific antibody (should be validated).

Response: We apologize for the confusion. The sTREM2-specific antibody was generated by immunizing rabbits with a peptide near the C-terminus of sTREM2 (Ac-CESFEDAHVEH). The antiserum was purified using the immunogen peptide and counter-screened with full-length TREM2. We described the details in the revised manuscript (**Page 10, lines 393-399**). As suggested by the reviewer, we provided evidence to show the specificity of this antibody. First, this antibody only recognizes sTREM2 but not full-length TREM2 by Western blotting (**Fig. S4B**). Second, we verified the specificity of this antibody in TREM2 knockout brain tissues (**Fig. S4C**). Third, pre-incubation with sTREM2 abolished the immunosignal of the antibody (**Fig. S4D**).

Fig. S4B-D

3) No antibody information for detecting Fc protein (Fig. 2E).

Response: We used anti-human IgG (Proteintech, 16402-1-AP) to detect the Fc protein. We described this in the revised manuscript (**Page 10, line 384**).

4) No detailed information on virus production (promoter, virus serotype, and so on).

Response: We apologize for missing information on virus production. We used AAV2/8 expressing human EGFP-2A-sTREM2-3×FLAG TREM2(1-171aa) (AAV-sTREM2) or EGFP (control AAV) under the control of the CAG promoter. The virus has been used previously to overexpress sTREM2 in the brain (Nat Commun, 2019, 10(1):1365). We described the detailed information in the Methods section (**Page 10, lines 401-409**).

5) No detail on how they quantify the images in vitro and in vivo.

Response: We described the details in the Methods section (**Page 12, lines 480-486**). The immunofluorescence images were quantified using ImageJ. The integrated pixel intensity of the region of interest (ROI) was quantified, and the background fluorescence intensity was subtracted. IHC images were quantified through ImageJ and the IHC Profiler plugin. The program counts the pixels and evaluates the percentage contributions of high positive, positive, low positive, and negative areas. The 'optical density score' was calculated as (percentage contribution of high positive*4 + percentage contribution of positive*3 + percentage contribution of low positive*2 + percentage contribution of negative*1)/100.

6) No information for the AD human brains and the brain areas that were used in the study.

Response: The brain tissues for immunostaining were obtained from the Emory Alzheimer's Disease Research Center (ADRC) Brain Bank. The diagnosis of AD was

confirmed by the presence of amyloid plaques and neurofibrillary tangles. The age at death of patients ranged between 52 and 70 years old, and the PMI of patients was 2.5 to 9 h. The area of the sections was the hippocampus. Informed consent was obtained from the subjects. The study was approved by the Ethics Committee. We added this information to the manuscript (**Page 8, lines 317-323**).

Reviewer #2:

In the present study the authors aim to demonstrate that sTREM2 interacts with transgelin-2 (TG2) in neurons. sTREM2 binding to TG2 resulted in TG2 activation and inhibition of the RhoA-ROCK-GSK3 β pathway, leading to reduced tau hyperphosphorylation in neurons. The authors also identified the minimal active sequence of sTREM2 responsible for the protective effect on tau phosphorylation (sTREM2 (77-89)). Overall, this is an interesting manuscript describing a novel potential sTREM2 interactor and a protective pathway in the context of tau pathology. While different in vitro and in vivo approaches were used to test the role of sTREM2 on tau phosphorylation, the work has some major weakness, such as the lack of critical experiments proving the specificity of sTREM2 and TG2 interaction in neurons, and the incomplete characterisation of TG2 expression/ TG2-sTREM2 interaction in other brain cells.

Response: We greatly appreciate the reviewer's constructive comments. We have revised the manuscript to fully address the reviewer's concerns.

Below, specific points of critique:

Major comments:

- The Authors state that “silencing of brain TREM2 exacerbates tau pathology^{15,16,17}, while overexpression of TREM2 ameliorates tau pathology¹”. This sentence is oversimplistic and does not consider the literature reporting a protective effect of TREM2 deficiency in models of tauopathy or showing a differential effect played by TREM2 in the context of β -amyloid and tau pathologies (PMID: 36219197; PMID: 29073081; PMID: 33675684; PMID: 30283031).

Response: We appreciate the reviewer's comments. As suggested by the reviewer, we provided a more comprehensive description of the effect of TREM2 on tau pathology, and included the references mentioned by the reviewer (**Page 2, lines 44-49, Page 6, lines 248-255**).

- Figure 1: Since different phosphorylation residues have been observed in Tauopathies (Wegmann et al., 2021 PMID: 33892381), authors should explain in the main text and discussion why they focused on S202 and S396 residues. Did the authors investigated whether sTREM2 treatment decrease tau phosphorylation also at other residues?

Response: We focused on the phosphorylation of S202 and S396 residues since they are implicated in the onset of AD (Wegmann et al., 2021 PMID: 33892381). We also

investigated whether sTREM2 treatment decreases tau phosphorylation at other residues, including T181, S404, and T231 (**Fig. 1A**). We found that the levels of p-Tau at the S202, S396, T181, and S404 residues were decreased after sTREM2 treatment, with S202 and S396 being the most dramatically altered residues. We have explained this in the main text (**Page 2, lines 72-74**) and discussed it in the Discussion section (**Page 7, lines 263-265**).

Fig. 1A

-Figure 1A: The authors generated a HEK293 cell line stably overexpressing GFP-Tau. They should provide immunofluorescence (IF) staining of GFP-Tau signal and show co-localization between GFP-Tau and p-S202 and p-S396 residues.

Response: We appreciate the reviewer's comments and provided immunofluorescence staining of GFP-Tau signal and show colocalization between GFP-Tau and p-S202 and p-S396 residues (**Fig. 1C, E, Fig. S1C, D**).

Fig. 1C, E

Fig. S1C, D

-Figure 1A-E: Is the control group (sTREM2= 0 nM) treated with heat-inactivated sTREM2? The use of heat-inactivated sTREM2 has been previously described in PMID: 28209725. To have the proper control, the authors should use heat-inactivated sTREM2 or at least add the same vehicle in which sTREM2 is purified.

Response: We appreciate the reviewer's comments. In Figure 1A-E, the control group was treated with Fc. We described this in the figure legends. Furthermore, we found that heat-inactivated sTREM2 lost the ability to inhibit tau phosphorylation (Fig. 1A-D, Fig. S1C).

Fig. 1A-D

Fig. S1C

-Fig. 1: Please specify in the M&M what sTREM2 has been used. How do the authors prove that sTREM2 is added at increasing concentrations to HEK293 cell line and to neuronal cultures? They should take advantage of a tagged-sTREM2 (e.g., HIS-tagged as used in Fig. 2) or measure sTREM2 levels by ELISA.

Response: We apologize for the confusion. In Figure 1, we used purified Fc-tagged sTREM2. Fc was used as control. We described the details in the Methods section (Pages 8-9, lines 331-336). The final concentrations of sTREM2 in the culture medium were determined using ELISA immediately after adding sTREM2 (Fig. S1B, Page 9, lines 334-336).

Fig. S1B

-Fig. 1D: Does sTREM2 addition to the media affect neuronal health? The authors should add a neuronal marker (e.g. MAP2, NeuN) and stain for propidium iodide/ Calcein to verify whether sTREM2 is affecting neuronal survival. The authors should also perform the same experiment using WT neurons (at least testing one dosage of sTREM2).

Response: We appreciate the reviewer's suggestions. Since the fixation step can not be performed after PI staining, we performed TUNEL staining to show the apoptotic neurons. We stained the neuronal marker MAP2 together with TUNEL and found that sTREM2 reduced apoptosis in primary neurons from Tau P301S mice (**Fig. S1F, G**). The anti-apoptotic effect of sTREM2 on WT neurons was not obvious due to the very low apoptosis rate in WT neurons. We also performed LDH release assay and found that sTREM2 decreased the release of LDH in Tau P301S neurons (**Fig. S1H**). These results suggest that sTREM2 is beneficial for neuronal survival.

Fig. S1F, G, H

-Fig. S1: There is no mention in the M&M of which anti-TREM2 antibody has been used to deplete the CM of sTREM2.

Response: We used TREM2 antibody from Proteintech (catalog number Ag26319) to delete sTREM2. We described this in the revised manuscript (**Page 9, line 339**).

-Figure 2: Why do the authors use SH-SY5Y cells instead of primary neurons from WT mice? WT primary neurons are more suitable to pull down neuronal cell membrane interactors and should be preferred. Also, Mass Spect on primary neurons from tau P301S transgenic mice would help clarifying whether sTREM2 and TG2 interaction is modulated during AD. If the authors cannot address this second point, they should at least discuss it in the manuscript.

Response: We appreciate the reviewer's comments. We used SH-SY5Y cells since it is easy to obtain enough cell membrane fractions for MS analysis. As suggested by the reviewer, we confirmed our result using primary WT neurons and neurons from tau P301S transgenic mice. We found the presence of TG2 in both WT and Tau P301S neuronal membrane proteins that bound to sTREM2 (**Fig. 2C, Table S2, S3**). We discussed this in the revised manuscript (**Page 7, lines 283-288**).

Fig. 2C

Table S2

Accession	Gene	Intensity (control)	Intensity (sTREM2)	sTREM2/control	Log2 (sTREM2/control)
Q9WVA4	Tagln2	9.76	15.62	58.32	5.87

Table S3

Accession	Gene	Intensity (control)	Intensity (sTREM2)	sTREM2/control	Log2 (sTREM2/control)
Q9WVA4	Tagln2	8.71	15.47	108.03	6.76

- Fig. 2: The authors should verify the purification of Fc-tagged sTREM2 by western blot and provide the purity of cell membrane fraction (enrichment in membrane proteins and absence of nuclear proteins).

Response: As suggested by the reviewer, we verified the purification of Fc-tagged sTREM2 by Coomassie blue staining and Western blotting (Fig. S1A). We also determined the purity of the cell membrane fraction by Western blotting using the membrane fraction marker ATP1A and nuclear protein marker KDM1 (Fig. S4A).

Fig. S1A

Fig. S4A

-Table S1: LC MS/MS analysis of sTREM2 interactors found using the cell membrane fraction extracted from SH-SY5Y cells revealed that TREM2 protein interacts with sTREM2 with the highest affinity ($\text{Log}_2(\text{FcsTREM2}/\text{Fc}) = 10.4757$). How do the authors explain the expression of TREM2 protein in SH-SY5Y neuronal cells? This is a particularly crucial point since no one has previously reported TREM2 protein expression in neurons.

Response: sTREM2 was not expressed in SH-SY5Y cells. As indicated in Fig. 2A, we incubated purified Fc-tagged sTREM2 with membrane fractions of SH-SY5Y cells, and then performed LC-MS/MS. TG2, but not sTREM2, is expressed in SH-SY5Y cells. We apologize for the confusion and clarified this in the revised table legend (**Table S1**).

-Fig. 2C and D: What brain area is represented in the images? The authors should add lower magnification images of the brain area analysed. Also, adding a neuronal marker other than NeuN (e.g. Map2) would help clarifying where sTREM2 and TG2 localize on neurons since NeuN is a DNA-binding protein and it is not expressed at the neuronal membrane. In this regard, could the authors show whether sTREM2 and TG2 are present on the cell soma only (as it appears from the confocal images in Fig 2C and D) or are expressed on axons/ branches too?

Response: The hippocampus was represented in the images. We indicated this in the figure legend. We also added lower magnification images of the brain to show the localization of sTREM2 on neurons (**new Fig. 2D, upper panel**). We also co-stained sTREM2 with MAP2 to clarify the localization of sTREM2 and TG2 on neurons (**new Fig. 2D**). We found that sTREM2 and TG2 are present both in the cell soma and branches of neurons.

new Fig. 2D

The signal of TG2 in Fig. 2D cannot be clearly distinguished from the background. Confocal images showing TG2 expression in mouse brains with a better resolution and lower background should be provided. Also, 3D reconstruction of confocal z-stack would help strengthening the authors' finding.

Response: As suggested by the reviewer, we provided clearer confocal images showing the localization of TG2 in the mouse brain (Fig. 2F). We also provided the 3D reconstruction of the confocal z-stack (Fig. 2E).

Fig. 2F

Fig. 2E

- TG2 is mostly expressed by smooth muscle cells and in the brain, RNA-seq datasets show that TAGLN2 gene is expressed at high levels by microglia/endothelial cells rather than neurons (<http://www.brainrnaseq.org/>). The authors should show if TG2-sTREM2 interaction occurs in glial cells by using microglial (IBA1, P2Y12 receptor) and astrocytic (GFAP) markers. They should also discuss the meaning of this interaction in both glial cells and neurons in the context of tau pathology.

Response: We appreciate the reviewer's comments. Although the mRNA level of TG2 in neurons is relatively low as detected by RNAseq (<https://www.brainrnaseq.org/>), a high-resolution mass spectrometry-based proteomics for in-depth analysis of the major brain regions and cell types identified that TG2 is evenly expressed in neurons and glial cells (Nature Neuroscience, 2015, 18:1819-1831). As suggested by the reviewer, we performed multiplex immunofluorescence to show the TG2-sTREM2 interaction in glial cells by using microglial (IBA1) and astrocytic (GFAP) markers (**Fig. S4E**). We found that sTREM2 also co-localizes with TG2 on microglia and astrocytes. However, it remains unknown whether the TG2-sTREM2 interaction on glial cells affects tau pathology. We discussed this in the Discussion section (**Page 7, lines 280-283**).

Fig. S4E

-Figure S7: Bands for TG2 protein and p-RhoA S188 by WB in AD patients are not very convincing. The authors should show clearer TG2 and p-RhoA S188 representative bands.

Response: We provided clearer TG2 and p-RhoA S188 representative bands in the new **Fig. S8A**.

new Fig. S8A

-S7: Could the authors perform immunofluorescence staining of TG2 together with a neuronal marker in AD versus control patients?

Response: We appreciate the reviewer's comments. As suggested, we performed immunofluorescence staining of TG2 together with MAP2 in brain sections from AD patients and control subjects (**new Fig. S8C**).

new Fig. S8C

-Figure 3E: The authors should provide WB and/or IF staining to confirm TG2 silencing upon shRNA treatment.

Response: In Figure 3E, we provided WB images confirming TG2 silencing upon shRNA treatment (4th pane in Fig. 3E).

Fig. 3E

-Fig. S8: The authors should add an image of hippocampus from mice injected with control AAVs in parallel to the AAV-EGFP-sTREM2 injected ones.

Response: We appreciate the reviewer's comments. As suggested, we provided an image of the hippocampus from mice injected with control AAVs in parallel to the AAV-EGFP-sTREM2 injected ones (*new Fig. S9A*).

new Fig. S9A

-Fig. 5 and 6: the authors should provide IF staining for inhibitory and excitatory synapses (e.g. PSD-95 and VGLUT-1) to test whether both compartments are involved.

Response: This is a very interesting point. We followed the reviewer's comments and performed IF staining for inhibitory and excitatory synapses. The results showed that sTREM2 rescued the loss of both inhibitory and excitatory synapses in the hippocampus of Tau P301S mice (**Fig. S9D-G, S10C-F**).

Fig. S9D-G

Fig. S10C-F

Figure 5D: In the EM images, it is not clear what the red stars are pointing. Could the authors provide images with a better resolution? Since it looks like that the red stars cover the synaptic sites, replacing them with smaller stars/arrows would help the reader understanding the figure better.

Response: We appreciate the reviewer's comments. We provided images with a better resolution and replaced the red starts with arrows (**Fig. 5D**).

Fig. 5D

Fig. S9: As previously pointed out, the authors should provide lower magnification images to have a better picture of TG2 expression in the hippocampal region. They should also add a neuronal marker to show TG2 localization on neurons.

Response: We appreciate the reviewer's comments. We provided lower magnification images to show the expression of TG2 in the hippocampal region. We also added the neuronal marker MAP2 to show TG2 localization on neurons (**new Fig. S10B**).

New Fig. S10B

Fig. 8A and S13B: Representative images of FITC fluorescence in the brain of tau P301S mice injected i.p. with Tat-sTREM2 (77-89) or Tat-sTREM2 (89-77) are shown. The authors should stain these sections with an anti-TG2 antibody and with a neuronal marker (e.g., NeuN, Map2). This experiment will help understanding whether the FITC-peptide localizes at specific sites of the cell and colocalizes/ binds TG2 as observed for sTREM2.

Response: As suggested, we stained the sections with an anti-TG2 antibody and anti-MAP2 antibody to show the colocalization of the peptide with TG2 on the neuronal membrane (Fig. S13D).

Fig. S13D

Minor comments:

- HEK293 and SH-SY5Y cell culture and treatment should be more detailed in the M&M section.

Response: We appreciate the reviewer's comments and described the details of HEK293 and SH-SY5Y cell culture and treatment in the Methods section (Page 8, lines 325-358).

- Please provide the expanded definition for SH-SY5Y cells.

Response: We provided the expanded definition of SH-SY5Y cells. SH-SY5Y cells are a subclone of a neuroblastoma cell line called SK-N-SH (SH-SY). The SH-SY cell line was subcloned as SH-SY5, which was subcloned for the third time to produce the SH-SY5Y line. We obtained SH-SY5Y cells from the American Tissue Culture Collection (ATCC, cat. CRL-2266). We described this in the Methods section (Page 8, lines 326-327).

- Fig.1D –3D: the authors should provide more information about neuronal culture and treatment in the M&M section.

Response: As suggested by the reviewer, we provided detailed information on neuronal culture and treatment in the Methods section (Pages 9-10, lines 360-378).

- Fig.1 – S4 – 7E: It should be clarified for how long the cells are incubated with sTREM2.

Response: In Figs. 1, S4 (new S6), and 7E, the cells were incubated with sTREM2 for 24 h. We described this in the Materials and Methods section (**Pages 8-9, lines 325-358**) and in each figure legend.

- Figure: S1, S2, S4, S5: Letters in the figure legends are missing.

Response: We added the letters in the figure legends.

- Figure 4B,4D,4F: please provide details about dosages, time of treatment and specify the name of RhoA activator.

Response: As suggested by the reviewer, we added the dosages, time of treatment, and name of RhoA activator (Rho Activator I, Cytoskeleton, # CN01) (**Page 9, lines 353-355**).

- Figure 5: The authors should clarify in the main text or in the figure legend whether mice were analysed at 7 months of age for all the tests performed.

Response: Yes, all the mice were analyzed at 7 months of age for the behavioral experiment. We clarified this in the revised manuscript (**Page 4, line 162**).

- Fig. S8: Please, specify in the fig. legend at what age tau P301S mice were injected with AAV-EGFP-sTREM2.

Response: Three-month-old tau P301S mice were injected with AAV-EGFP-sTREM2. We added this information to the figure legend of the **new Fig. S9A**.

- Fig. S13: Please, specify in the fig. legend which brain area is represented.

Response: As suggested, we provided images in the hippocampal region. We added this information to the figure legend (**Fig. S13A**). We also added images of other brain areas in **Fig. S13C**.

- Fig. 8G and 5F: statistical significances are missing in the graphs.

Response: We added statistical significance to **Figs. 8G and 5F**.

REVIEWER COMMENTS

Reviewer #1 (Remarks to the Author):

The authors have adequately addressed my concerns and no further question from me.

Reviewer #2 (Remarks to the Author):

In the revised version of the manuscript, the authors addressed most of the reviewers' comments and added new experiments. However, I still have concerns about the specificity of some of the staining performed to prove the expression of TG2 on neurons, astrocytes and microglia *in vivo* and its colocalization with sTREM2.

Below some comments:

Figure 1: Since different phosphorylation residues have been observed in Tauopathies (Wegmann et al., 2021 PMID: 33892381), authors should explain in the main text and discussion why they focused on S202 and S396 residues. Did the authors investigate whether sTREM2 treatment decrease tau phosphorylation also at other residues?

Response: We focused on the phosphorylation of S202 and S396 residues since they are implicated in the onset of AD (Wegmann et al., 2021 PMID: 33892381). We also investigated whether sTREM2 treatment decreases tau phosphorylation at other residues, including T181, S404, and T231 (Fig. 1A). We found that the levels of p-Tau at the S202, S396, T181, and S404 residues were decreased after sTREM2 treatment, with S202 and S396 being the most dramatically altered residues. We have explained this in the main text (Page 2, lines 72-74) and discussed it in the Discussion section (Page 7, lines 263-265).

The authors should discuss deeper the fact that despite tau-phosphorylation on T231 is implicated in neurodegenerative pathologies, sTREM2 mediated inhibition of tau-phosphorylation is not observed at this residue. This implies that pathways unrelated to sTREM2 are involved in the phosphorylation of other residues, and this should be mentioned.

Figure 2: Why do the authors use SH-SY5Y cells instead of primary neurons from WT mice? WT primary neurons are more suitable to pull down neuronal cell membrane interactors and should be preferred. Also, Mass Spect on primary neurons from tau P301S transgenic mice would help clarifying whether sTREM2 and TG2 interaction is modulated during AD. If the authors cannot address this second point, they should at least discuss it in the manuscript.

Response: We appreciate the reviewer's comments. We used SH-SY5Y cells since it is easy to obtain enough cell membrane fractions for MS analysis. As suggested by the reviewer, we confirmed our result using primary WT neurons and neurons from tau P301S transgenic mice. We found the presence of TG2

in both WT and Tau P301S neuronal membrane proteins that bound to sTREM2 (Fig. 2C, Table S2, S3). We discussed this in the revised manuscript (Page 7, lines 283-288).

The authors should update the cartoon in Fig.2a adding neurons label.

Fig. S1: There is no mention in the M&M of which anti-TREM2 antibody has been used to deplete the CM of sTREM2.

Response: We used TREM2 antibody from Proteintech (catalog number Ag26319) to delete sTREM2. We described this in the revised manuscript (Page 9, line 339).

Ag26319 from Proteintech is a Trem2 fusion protein, did the authors refer to any published article to use this fusion protein? Please add the reference in the text. Why do the authors use a fusion protein to deplete sTREM2 and not a commercially available antibody?

Table S1: LC MS/MS analysis of sTREM2 interactors found using the cell membrane fraction extracted from SH-SY5Y cells revealed that TREM2 protein interacts with sTREM2 with the highest affinity ($\text{Log}_2(\text{FcsTREM2}/\text{Fc}) = 10.4757$). How do the authors explain the expression of TREM2 protein in SH-SY5Y neuronal cells? This is a particularly crucial point since no one has previously reported TREM2 protein expression in neurons.

Response: sTREM2 was not expressed in SH-SY5Y cells. As indicated in Fig. 2A, we incubated purified Fc-tagged sTREM2 with membrane fractions of SH-SY5Y cells, and then performed LC-MS/MS. TG2, but not sTREM2, is expressed in SH-SY5Y cells. We apologize for the confusion and clarified this in the revised table legend (Table S1).

The authors did not answer to the question. How do the authors explain the expression of TREM2 protein (Q9NZC2 TREM2) (not sTREM2) in SH-SY5Y neuronal cells?

Fig. 2C and D: What brain area is represented in the images? The authors should add lower magnification images of the brain area analysed. Also, adding a neuronal marker other than NeuN (e.g. Map2) would help clarifying where sTREM2 and TG2 localize on neurons since NeuN is a DNA-binding protein and it is not expressed at the neuronal membrane. In this regard, could the authors show whether sTREM2 and TG2 are present on the cell soma only (as it appears from the confocal images in Fig 2C and D) or are expressed on axons/ branches too?

Response: The hippocampus was represented in the images. We indicated this in the figure legend. We also added lower magnification images of the brain to show the localization of sTREM2 on neurons (new Fig. 2D, upper panel). We also co-stained sTREM2 with MAP2 to clarify the localization of sTREM2 and TG2 on neurons (new Fig. 2D). We found that sTREM2 and TG2 are present both in the cell soma and branches of neurons.

MAP2 staining in the hippocampus is unconvincing (<https://www.proteinatlas.org/ENSG00000078018-MAP2/tissue/hippocampus>).

Also, both Reviewer1 and 2 pointed out that TG2 is highly expressed by microglia, endothelial cells and OPC (as reported by brain RNAseq database). Why in the lower magnification images TG2 seem to be expressed only on neurons?

The signal of TG2 in Fig. 2D cannot be clearly distinguished from the background. Confocal images showing TG2 expression in mouse brains with a better resolution and lower background should be provided. Also, 3D reconstruction of confocal z-stack would help strengthening the authors' finding.

Response: As suggested by the reviewer, we provided clearer confocal images showing the localization of TG2 in the mouse brain (Fig. 2F). We also provided the 3D reconstruction of the confocal z-stack (Fig. 2E).

The 3D reconstructions images are still not very convincing. Since colocalization of sTREM2 and TG2 is central in the manuscript, more higher magnification images of neurons need to be shown to prove this interaction (not only of the neuronal soma).

TG2 is mostly expressed by smooth muscle cells and in the brain, RNA-seq datasets show that TAGLN2 gene is expressed at high levels by microglia/endothelial cells rather than neurons (<http://www.brainrnaseq.org/>). The authors should show if TG2- sTREM2 interaction occurs in glial cells by using microglial (IBA1, P2Y12 receptor) and astrocytic (GFAP) markers. They should also discuss the meaning of this interaction in both glial cells and neurons in the context of tau pathology.

Response: We appreciate the reviewer's comments. Although the mRNA level of TG2 in neurons is relatively low as detected by RNAseq (<https://www.brainrnaseq.org/>), a high-resolution mass spectrometry-based proteomics for in-depth analysis of the major brain regions and cell types identified that TG2 is evenly expressed in neurons and glial cells (Nature Neuroscience, 2015, 18:1819-1831). As suggested by the reviewer, we performed multiplex immunofluorescence to show the TG2-sTREM2 interaction in glial cells by using microglial (IBA1) and astrocytic (GFAP) markers (Fig. S4E). We found that sTREM2 also co-localizes with TG2 on microglia and astrocytes. However, it remains unknown whether the TG2-sTREM2 interaction on glial cells affects tau pathology. We discussed this in the Discussion section (Page 7, lines 280-283).

Did the authors stain GFAP and Iba1 in P301S mice or in AD brains only? They should add GFAP/ Iba1/ sTREM2 and TG2 staining in P301S mice too.

REVIEWER COMMENTS

Reviewer #1 (Remarks to the Author):

The authors have adequately addressed my concerns and no further question from me.

Reviewer #2 (Remarks to the Author):

In the revised version of the manuscript, the authors addressed most of the reviewers' comments and added new experiments. However, I still have concerns about the specificity of some of the staining performed to prove the expression of TG2 on neurons, astrocytes and microglia in vivo and its colocalization with sTREM2. Below some comments:

Figure 1: Since different phosphorylation residues have been observed in Tauopathies (Wegmann et al., 2021 PMID: 33892381), authors should explain in the main text and discussion why they focused on S202 and S396 residues. Did the authors investigate whether sTREM2 treatment decrease tau phosphorylation also at other residues?

Response: We focused on the phosphorylation of S202 and S396 residues since they are implicated in the onset of AD (Wegmann et al., 2021 PMID: 33892381). We also investigated whether sTREM2 treatment decreases tau phosphorylation at other residues, including T181, S404, and T231 (Fig. 1A). We found that the levels of p-Tau at the S202, S396, T181, and S404 residues were decreased after sTREM2 treatment, with S202 and S396 being the most dramatically altered residues. We have explained this in the main text (Page 2, lines 72-74) and discussed it in the Discussion section (Page 7, lines 263-265).

The authors should discuss deeper the fact that despite tau-phosphorylation on T231 is implicated in neurodegenerative pathologies, sTREM2 mediated inhibition of tau-phosphorylation is not observed at this residue. This implies that pathways unrelated to sTREM2 are involved in the phosphorylation of other residues, and this should be mentioned.

Response: We appreciate the reviewer's comments and discussed this in the Discussion section. "Tau phosphorylation at S202, S396, S404, T181, and T231 residues is involved in the pathogenesis of AD^{34,35,36,37}. We found that sTREM2 potently inhibits the phosphorylation of tau at the S202, S396, T181, and S404 residues, but not at T231. These results indicate that pathways unrelated to sTREM2 are involved in tau phosphorylation at T231" (Page 7, lines 261-264).

Figure 2: Why do the authors use SH-SY5Y cells instead of primary neurons from WT mice? WT primary neurons are more suitable to pull down neuronal cell membrane interactors and should be preferred. Also, Mass Spect on primary neurons from tau P301S transgenic mice would help clarifying whether sTREM2 and TG2 interaction is modulated during AD. If the authors cannot address this second point, they should at least discuss it in the manuscript.

Response: We appreciate the reviewer's comments. We used SH-SY5Y cells since it is easy to obtain enough cell membrane fractions for MS analysis. As suggested by the reviewer, we confirmed our result using primary WT neurons and neurons from tau P301S transgenic mice. We found the presence of TG2 in both WT and Tau P301S neuronal membrane proteins that bound to sTREM2 (Fig. 2C, Table S2, S3). We discussed this in the revised manuscript (Page 7, lines 283-288).

The authors should update the cartoon in Fig.2a adding neurons label.

Response: Thank you for the reminder. We have updated the cartoon in Fig. 2A.

Fig. 2A

Fig. S1: There is no mention in the M&M of which anti-TREM2 antibody has been used to deplete the CM of sTREM2.

Response: We used TREM2 antibody from Proteintech (catalog number Ag26319) to delete sTREM2. We described this in the revised manuscript (Page 9, line 339).

Ag26319 from Proteintech is a Trem2 fusion protein, did the authors refer to any published article to use this fusion protein? Please add the reference in the text. Why do the authors use a fusion protein to deplete sTREM2 and not a commercially available antibody?

Response: We are very sorry for the typo. We used the TREM2 antibody from Proteintech (catalog number 27599-1-AP) to delete sTREM2. The fusion protein (Catalog number #Ag26319) from Proteintech was the immunogen to generate this antibody. We accidentally confused the two catalog numbers. ELISA analysis confirmed that this antibody efficiently depleted sTREM2 from the conditioned medium (Fig. S2C).

Table S1: LC MS/MS analysis of sTREM2 interactors found using the cell membrane fraction extracted from SH-SY5Y cells revealed that TREM2 protein interacts with sTREM2 with the highest affinity ($\text{Log}_2(\text{FcsTREM2/Fc}) = 10.4757$). How do the authors explain the expression of

TREM2 protein in SH-SY5Y neuronal cells? This is a particularly crucial point since no one has previously reported TREM2 protein expression in neurons.

Response: sTREM2 was not expressed in SH-SY5Y cells. As indicated in Fig. 2A, we incubated purified Fc-tagged sTREM2 with membrane fractions of SH-SY5Y cells, and then performed LC-MS/MS. TG2, but not sTREM2, is expressed in SH-SY5Y cells. We apologize for the confusion and clarified this in the revised table legend (Table S1).

The authors did not answer to the question. How do the authors explain the expression of TREM2 protein (Q9NZC2 TREM2) (not sTREM2) in SH-SY5Y neuronal cells?

Response: We are sorry for the confusion. TREM2 is NOT expressed in SH-SY5Y cells. In our experiments, purified Fc-tagged sTREM2 was attached to Protein A beads and incubated with membrane fractions of SH-SY5Y cells. All the proteins attached to the beads including Fc-sTREM2 were subjected to mass spectrometry. The mass spectrometry data generated by TripleTOF5600 were searched through ProteinPilot (V4.5) using the database retrieval algorithm Paragon. Thus, the peptide sequence of sTREM2 detected by mass spectrometry was incorrectly annotated as TREM2 (Q9NZC2). To avoid the confusion, we deleted TREM2 from the list and explained this issue in the Table legend.

Fig. 2C and D: What brain area is represented in the images? The authors should add lower magnification images of the brain area analysed. Also, adding a neuronal marker other than NeuN (e.g. Map2) would help clarifying where sTREM2 and TG2 localize on neurons since NeuN is a DNA-binding protein and it is not expressed at the neuronal membrane. In this regard, could the authors show whether sTREM2 and TG2 are present on the cell soma only (as it appears from the confocal images in Fig 2C and D) or are expressed on axons/ branches too?

Response: The hippocampus was represented in the images. We indicated this in the figure legend. We also added lower magnification images of the brain to show the localization of sTREM2 on neurons (new Fig. 2D, upper panel). We also co-stained sTREM2 with MAP2 to clarify the localization of sTREM2 and TG2 on neurons (new Fig. 2D). We found that sTREM2 and TG2 are present both in the cell soma and branches of neurons.

MAP2 staining in the hippocampus is unconvincing

(<https://www.proteinatlas.org/ENSG00000078018-MAP2/tissue/hippocampus>). Also, both Reviewer1 and 2 pointed out that TG2 is highly expressed by microglia, endothelial cells and OPC (as reported by brain RNAseq database). Why in the lower magnification images TG2 seem to be expressed only on neurons?

Response: We provided more representative images in new Fig. 2D. The pattern of MAP2 staining is consistent with that from the Human Protein Atlas (New Fig. 2D). In the lower magnification images, TG2 is also expressed in other types of cells in addition to neurons (arrows). To more clearly show this, we also provided higher magnification images of TG2 co-staining with the microglia marker Iba-1, endothelial marker CD31, and OPC marker (Fig. S4E).

New Fig. 2D

MAP2 staining in the Human Protein Atlas

Fig. S4E

The signal of TG2 in Fig. 2D cannot be clearly distinguished from the background. Confocal images showing TG2 expression in mouse brains with a better resolution and lower background should be provided. Also, 3D reconstruction of confocal z-stack would help strengthening the authors' finding.

Response: As suggested by the reviewer, we provided clearer confocal images showing the localization of TG2 in the mouse brain (Fig. 2F). We also provided the 3D reconstruction of the confocal z-stack (Fig. 2E).

The 3D reconstructions images are still not very convincing. Since colocalization of sTREM2 and TG2 is central in the manuscript, more higher magnification images of neurons need to be shown to prove this interaction (not only of the neuronal soma).

Response: As suggested by the reviewer, we provided higher magnification images of neurons showing the colocalization between TG2 and sTREM2 (**Fig. 2E, F**). We also analyzed the intensity traces (offset white line) of MAP2, TG2, sTREM2, and DAPI. We found that TG2 and sTREM2 had similar intensity traces, especially at the "i" and "ii" positions. The arrows indicate the colocalization in branches.

Fig. 2E, F

F

TG2 is mostly expressed by smooth muscle cells and in the brain, RNA-seq datasets show that TAGLN2 gene is expressed at high levels by microglia/endothelial cells rather than neurons (<http://www.brainrnaseq.org/>). The authors should show if TG2- sTREM2 interaction occurs in glial cells by using microglial (IBA1, P2Y12 receptor) and astrocytic (GFAP) markers. They should also discuss the meaning of this interaction in both glial cells and neurons in the context of tau pathology.

Response: We appreciate the reviewer's comments. Although the mRNA level of TG2 in neurons is relatively low as detected by RNAseq (<https://www.brainrnaseq.org/>), a high-resolution mass spectrometry-based proteomics for in-depth analysis of the major brain regions and cell types identified that TG2 is evenly expressed in neurons and glial cells (Nature Neuroscience, 2015, 18:1819-1831). As suggested by the reviewer, we performed multiplex immunofluorescence to show the TG2-sTREM2 interaction in glial cells by using microglial (IBA1) and astrocytic (GFAP) markers (Fig. S4E). We found that sTREM2 also co-localizes with TG2 on microglia and astrocytes. However, it remains unknown whether the TG2-sTREM2 interaction on glial cells affects tau pathology. We discussed this in the Discussion section (Page 7, lines 280-283).

Did the authors stain GFAP and Iba1 in P301S mice or in AD brains only? They should add GFAP/ Iba1/ sTREM2 and TG2 staining in P301S mice too.

Response: As suggested by the reviewer, we have added GFAP/Iba1/sTREM2 and TG2 staining in Tau P301S mice in Fig. S4F.

Fig. S4F

REVIEWERS' COMMENTS

Reviewer #2 (Remarks to the Author):

The authors have addressed the majority of my concerns and the quality of the manuscript has improved.

I have only one last minor comment:

Labels in Fig. 2B showing MassSpect analysis are not readable. Please, increase the resolution.

Reviewer #2 (Remarks to the Author):

The authors have addressed the majority of my concerns and the quality of the manuscript has improved.

I have only one last minor comment:

Labels in Fig. 2B showing MassSpect analysis are not readable. Please, increase the resolution.

Response: As suggested by the reviewer, we have increased the resolution of Fig. 2B.